# Parametric covariance dynamics for the nonlinear diffusive Burgers equation

Olivier Pannekoucke[1], Marc Bocquet[2], and Richard Ménard[3]

[1] INPT-ENM, CNRM UMR 3589, Météo-France/CNRS, CERFACS, Toulouse, France.
[2] CEREA, joint laboratory École des Ponts ParisTech and EDF R&D, Université Paris-Est, Champs-sur-Marne, France
[3] ARQI/Air Quality Research Division, Environment and Climate Change Canada, Dorval (Québec), Canada.

*Correspondence to:* O. Pannekoucke (olivier.pannekoucke@meteo.fr)

**Abstract.** The parametric Kalman filter (PKF) is a computationally efficient alternative method to the ensemble Kalman filter (EnKF). The PKF relies on an approximation of the error covariance matrix by a covariance model with a space-time evolving set of parameters. This study extends the PKF to nonlinear dynamics using the diffusive Burgers equation as an application, focusing on the forecast step of the assimilation cycle. The covariance model considered is based on the diffusion equation, with the diffusion tensor and the error variance as evolving parameters. An analytical derivation of the parameter dynamics highlights a closure issue. Therefore, a closure model is proposed based on the kurtosis of the local correlation functions. Numerical experiments compare the PKF forecast with the statistics obtained from a large ensemble of nonlinear forecasts. These experiments strengthen the closure model and demonstrate the ability of the PKF to reproduce the tangent-linear covariance dynamics, at a low numerical cost.

## 1 Introduction

Covariance functions in geophysical flows are known to evolve in both time and space (e.g., Bouttier, 1993; Snyder et al., 2003). Yet, an accurate solution of the covariance dynamics is one of the major challenges in data assimilation and probabilistic forecasting. The Monte Carlo method, which is the most common approach, addresses nonlinear dynamics and is computationally efficient with parallel computers. However, it suffers from non-uniform sampling noise, which is a function of the true signal covariance.

Another route can be investigated that relies on analytical derivation of covariance tensor dynamics (Cohn, 1993), which has inspired application in chemical transport model (Ménard et al., 2000). Despite the theoretical interest resulting from the analytical derivation of covariance dynamics, it is still difficult to take advantage of this formulation in real applications. Moreover, the system presents closure problem for the diffusive error dynamics. An hybrid approach that mixes the Monte Carlo method based on an ensemble and an approximate propagation of the correlations by a surrogate model has also been proposed (Bocquet, 2016).

An intermediate formulation, between the approximation by an ensemble and the theoretical formulation by analytic derivation, has been recently introduced by Pannekoucke et al. (2016) who proposed to approximate the forecast error covariance matrix by a parametric covariance matrix, from which the dynamics of parameters stand for the dynamics of the full covariance matrix. This formulation, called the Parametric Kalman Filter (PKF), has been illustrated on a linear advection-diffusion equation similarly to the equations encountered in chemical transport model. As defined for general parametric covariance model, the PKF has been illustrated for the particular case where the covariance model is based on the diffusion equation (Weaver and Courtier, 2001). Hence, the error covariance matrix is reduced to the knowledge of its variance field and its local diffusion tensor field. The time evolution all along the forecast and the analysis steps of the data assimilation process is expressed in terms of variance and local diffusion tensor evolution.

As mentioned earlier, the PKF formulation has been tested so far on a linear dynamics. It is thus interesting for more general applications to consider extension to a nonlinear setting. The goal of the present work is to formulate and illustrate the forecast step of the PKF for the nonlinear dynamics given by Burgers equation. Burgers equation is a nonlinear advection-diffusion model that usually involves one variable in a one-dimensional space – the $u-$wind. It is one of the simplest equations that display important features of geophysical interest, such as advection, frontogenesis and 1D turbulence (Burgers (1974); Hopf (1950); el Malek and El-Mansi (2000) and references therein). Burgers equation has been used in several data assimilation studies to examine the effect of nonlinearity error propagation and in Kalman filtering methods (Cohn, 1993; Ménard, 1994; Verlaan and Heemink, 2001), in maximum likelihood ensemble filtering (Zupanski et al., 2008), in adjoint methods (Apte et al., 2010), in model error estimation using 4D-Var (Lakshmivarahan et al., 2013), and in 4DEnVar and localization (Desroziers et al., 2014, 2016).

However, preliminary numerical tests have shown that the treatment of the physical diffusion as proposed in Pannekoucke et al. (2016), and deduced from analytical solution, was not able to reproduce the complexity of the Burgers dynamics. Hence, we need to develop a higher order representation of the PKF equation for the physical diffusion process.

In section 2 the parametric formulation based on the covariance modeling with the diffusion equation is first recalled, and we specify the methodology for developing the parametric dynamics under a systematic treatment. This method is then applied to the Burgers equation, in section 3, taking advantage of the operator splitting. In section 4, numerical simulations are conducted to illustrate the ability of the parametric dynamics to reproduce the main features of the true covariance dynamics emerging from a forecast Monte Carlo experiment. The conclusions are given in section 5.

## 2 Parametric formulation of covariance dynamics

### 2.1 Background on the uncertainty propagation and covariance dynamics

Geophysical flow dynamics can be represented as a nonlinear system of the form

$$\begin{cases} \partial_t u = \mathcal{M}(u), \\ u(\boldsymbol{x},0) = u^0(\boldsymbol{x}), \end{cases} \tag{1}$$

that describes the time evolution of a state function $u$ and where a unique solution is assumed to exist for any initial condition $u^0$ within an appropriate set.

Due to the lack of precise knowledge of the initial condition, $u^0$ is generally represented as a random state $u^0 = \overline{u^0} + \varepsilon^0$ where $\varepsilon^0$ is a centered Gaussian random field characterized by its two-point covariance function $P^0(\boldsymbol{x},\boldsymbol{y}) = \overline{\varepsilon^0(\boldsymbol{x})\varepsilon^0(\boldsymbol{y})}$, where $\overline{(\cdot)}$ stands for the expectation operator. The covariance function can be described by its variance field $V^0$, where $V_{\boldsymbol{x}}^0 = P^0(\boldsymbol{x},\boldsymbol{x})$ denotes the variance at position $\boldsymbol{x}$ and by its error correlation function $\rho^0(\boldsymbol{x},\boldsymbol{y}) = P^0(\boldsymbol{x},\boldsymbol{y})/\sigma_{\boldsymbol{x}}^0\sigma_{\boldsymbol{y}}^0$ where $\sigma_{\boldsymbol{x}}^0 = \sqrt{V_{\boldsymbol{x}}^0}$ and $\sigma_{\boldsymbol{y}}^0 = \sqrt{V_{\boldsymbol{y}}^0}$ are the standard deviations at point $\boldsymbol{x}$ and $\boldsymbol{y}$, respectively. When a first order Taylor expansion accurately approximates the error dynamics, then the tangent-linear dynamics of the initial error $\varepsilon^0$

$$\begin{cases} \partial_t \varepsilon = M\varepsilon, \\ \varepsilon(\boldsymbol{x},0) = \varepsilon^0(\boldsymbol{x}), \end{cases} \tag{2}$$

makes evolve the error about the mean state $\overline{u}$. $M = \partial_u \mathcal{M}_{|\overline{u}}$ is the tangent-linear dynamics along the nonlinear trajectory, $\overline{u}$, solution of Eq. (1) starting from the initial condition $\overline{u^0}$.

The two-point covariance function $\overline{\varepsilon(\boldsymbol{x},t)\varepsilon(\boldsymbol{y},t)}$ of the error field $\varepsilon$ at a given time $t$ defines the covariance function $P(\boldsymbol{x},\boldsymbol{y},t)$. Thereafter, the covariance function is computed as a covariance matrix: when space is discretized, with the grid-point positions denoted by $\boldsymbol{x}_i$, the restriction of the covariance function to the grid-point positions is the matrix $\mathbf{P}$ defined by $\mathbf{P}_{ij}(t) = P(\boldsymbol{x}_i,\boldsymbol{x}_j,t)$. With the discretized version of the tangent-linear model $M$ being now the matrix $\mathbf{M}$, the dynamics of the covariance matrix is then given by the forecast error covariance equation

$$\begin{cases} \frac{d}{dt}\mathbf{P} = \mathbf{M}\mathbf{P} + \mathbf{P}\mathbf{M}^{\mathrm{T}}, \\ \mathbf{P}(t=0) = \mathbf{P}^0, \end{cases} \tag{3}$$

where $\mathbf{M}^{\mathrm{T}}$ stands for the adjoint of the tangent-linear model $\mathbf{M}$. Thereafter, since the statistics depend from the time evolution, the explicit reference to the time $t$ is dropped, excepted for the initial time $t=0$ identified by superscript $(\cdot)^0$.

The numerical cost of solving Eq. (3) for high-dimensional dynamics is beyond supercomputer capacity. Different options have been considered in the literature to approximate the solution, among which one finds the Monte Carlo method employed in ensemble Kalman filter (Evensen, 1994).

The ensemble Kalman filter is a robust algorithm that applies to low order dynamical systems as well as to large dimension systems encountered in geophysical applications. The main difference for geophysical applications is that the covariance matrix

is closely related to the continuous covariance function, which may not be the case for all discrete low-order models. Thereafter, it is assumed that a discrete model results from the discretization of a continuous model, making a clear connection between the discrete and the continuous covariance representations. This offers simplifications in the following derivations. To that end, in what follows, the covariance function $P(\boldsymbol{x}, \boldsymbol{y})$ and its grid-point matrix representation $\mathbf{P}$ are considered as equivalent, and denoted by the matrix notation.

We now give details about another approximation, which relies on the continuum, namely the parametric formulation.

## 2.2 Parametric formulation of the covariance forecast dynamics

Pannekoucke et al. (2016) have proposed to parameterize the covariance matrix by the covariance model, and they have illustrated this approach by considering the covariance model based on the pseudo-time diffusion equation (Weaver and Courtier, 2001).

The diffusion covariance model factorizes the covariance matrix as

$$\mathbf{P}^{\text{diff.}} = \boldsymbol{\Sigma}\mathbf{L}\mathbf{L}^{\mathrm{T}}\boldsymbol{\Sigma}^{\mathrm{T}}, \tag{4}$$

where $\boldsymbol{\Sigma}$ denotes the diagonal matrix of standard deviation $\sqrt{\overline{\varepsilon^2}}$, and $\mathbf{L}$ is the propagator of the diffusion equation

$$\partial_\tau \alpha = \nabla \cdot (\boldsymbol{\nu}\nabla\alpha), \tag{5}$$

integrated from the pseudo-time $\tau$ from $\tau = 0$ to $\tau = 1/2$, thus giving $\mathbf{L} = e^{\frac{1}{2}\nabla \cdot (\boldsymbol{\nu}\nabla)}$. The pseudo-time diffusion equation is a recipe to build Gaussian random fields with Gaussian-like correlation functions. Note that, the pseudo-time $\tau$ has no link with the physical time $t$ of Eq. (1). In this formulation the variance field $V$ (deduced from $\boldsymbol{\Sigma}$) and the local diffusion tensor field ($\boldsymbol{\nu}$) are the only parameters to be dertermined.

Knowing the dynamics of the variance field $V$ and the local diffusion tensor field $\boldsymbol{\nu}$ provides a means to approximate the true covariance dynamics Eq. (3), where $\mathbf{P}$ would be replaced by the covariance model $\mathbf{P}^{\text{diff.}}$ parameterized by using the diffusion equation, Eq. (4). This constitutes the parametric formulation of the dynamics. The challenge is now to determine the dynamics of the two parameter fields.

The dynamics of the variance field $V = \overline{\varepsilon^2}$ can straightforwardly be obtained from the trend $\partial_t \varepsilon$ following

$$\partial_t V = 2\overline{\varepsilon\partial_t\varepsilon}. \tag{6}$$

However, the dynamics of the diffusion tensor is not as obvious to derive. A possible way to describe its dynamics is to consider some approximation that we will describe in the next section.

### 2.2.1 Approximate dynamics for the diffusion covariance model

The dynamical equations of the local diffusion can be obtained taking advantage of approximations used in data assimilation, for the estimation of the local diffusion tensor from ensemble data.

Following Pannekoucke and Massart (2008); Mirouze and Weaver (2010); Weaver and Mirouze (2013), the local diffusion tensor field can be deduced from the correlation functions, when assuming that the random error field is smooth. For a given position $\boldsymbol{x}$, the local Taylor expansion of the correlation function $\rho(\boldsymbol{x}, \boldsymbol{x} + \delta\boldsymbol{x})$ is related to the local correlation function in the form

$$\rho(\boldsymbol{x}, \boldsymbol{x} + \delta\boldsymbol{x}) \underset{0}{=} 1 - \frac{1}{2}||\delta\boldsymbol{x}||^2_{\boldsymbol{g_x}} + o(||\delta\boldsymbol{x}||^2), \tag{7}$$

where $\boldsymbol{g_x}$ denotes the local metric tensor at point $\boldsymbol{x}$, with $||\delta\boldsymbol{x}||^2_{\mathbf{E}} = \delta\boldsymbol{x}^{\mathrm{T}}\mathbf{E}\delta\boldsymbol{x}$. In this expression little $o$ means that for two functions, $f_1$ and $f_2$, $f_1 \underset{a}{=} o(f_2)$ is equivalent to $\lim_{x \to a} \frac{|f_1(x)|}{|f_2(x)|} = 0$. From Eq. (7) we can define a diffusion tensor at $\boldsymbol{x}$ by

$$\boldsymbol{\nu_x} = \frac{1}{2}\boldsymbol{g_x}^{-1}. \tag{8}$$

The importance of the metric tensor comes from its direct connection with the error field. In dimension one, the metric is the scalar $g_x = \overline{(\partial_x \tilde{\varepsilon}_x)^2}$ where $\tilde{\varepsilon}$ denotes the normalized error field $\tilde{\varepsilon}_x = \frac{\varepsilon_x}{\sigma_x}$ (see appendix A). It is meaningful to relate the metric to a typical scale of correlation, the so-called error correlation length-scale (Daley, 1991; Pannekoucke et al., 2008)

$$L_{\boldsymbol{x}} = 1/\sqrt{g_{\boldsymbol{x}}}. \tag{9}$$

In dimension two (three), the metric is a $2 \times 2$ ($3 \times 3$) matrix $\boldsymbol{g_x} = [g_{ij}(\boldsymbol{x})]$ given by

$$g_{ij} = \overline{\partial_{x^i}\tilde{\varepsilon}\partial_{x^j}\tilde{\varepsilon}}. \tag{10}$$

Consequently, an approximation for the dynamics of the parametric formulation based on the diffusion equation is given by

$$\partial_t V = 2\overline{\varepsilon \partial_t \varepsilon}, \tag{11a}$$

$$\partial_t g_{ij} = \partial_t \left( \overline{\partial_{x^i}\tilde{\varepsilon}\partial_{x^j}\tilde{\varepsilon}} \right). \tag{11b}$$

Equation (11), has the advantage that we should be able to compute the time evolution of covariances for any error dynamics. This will be illustrated with Burgers equation, that is a one dimensional dynamical model with nonlinear advection and diffusion processes similar to those of geophysical flows.

## 3  Dynamics of forecast error for Burgers model

Here, we consider the dynamics associated with Burgers equation

$$\partial_t u + u\partial_x u = \kappa\partial_x^2 u. \tag{12}$$

For any smooth function $u^0(x)$, there exists a unique solution $u(x,t)$ to Eq. (12) with the initial condition $u(x,0) = u^0(x)$. A particular initial condition is now considered where $u^0(x)$ is a sample of a smooth random field of mean field $\overline{u^0}(x)$. Hence, each sample $u^0$ is decomposed as $u^0(x) = \overline{u^0}(x) + \varepsilon^0(x)$, where $\varepsilon^0$ is a smooth random field. The dynamics of the

mean $\overline{u}(x,t)$ and of higher order statistical moments are obtained from the Reynolds equations. Similarly to Cohn (1993), the fluctuation-mean-flow dynamics deduced from the Reynolds equations is considered, in place of the more classical tangent-linear dynamics. Compared with the tangent-linear dynamics, fluctuation-mean-flow dynamics makes the mean flow depending on the fluctuation statistics evolution, which enlarge the tangent-linear setting. Note that the fluctuation-mean-flow interaction

leads to the *Gaussian second-order filter* (Jazwinski, 1970, sec. 9.3), and is important in *nonlinear Kalman-like filters* (Cohn, 1993). The next section presents the fluctuation-mean-flow dynamics, and how it is used to describe the time evolution Eq. (11) of the two-points error covariance parameters.

### 3.1    Derivation of the fluctuation-mean-flow dynamics for small error magnitudes

The random field $u$ can be decomposed into its ensemble-averaged and fluctuating parts $u = \overline{u} + \varepsilon$, where $\overline{u}(x,t) = \overline{u(x,t)}$

denotes the expectation of the random field $u$, and $\varepsilon = u - \overline{u}$ is a random field of zero mean. From this expansion, the mean flow dynamics is the ensemble average of the dynamics Eq. (12) reads

$$\partial_t \overline{u} + \overline{u}\partial_x \overline{u} = \kappa \partial_x^2 \overline{u} - \overline{\varepsilon \partial_x \varepsilon}. \tag{13a}$$

The dynamics of the fluctuation $\varepsilon$ is deduced from the difference between the full dynamics Eq. (12) and the mean flow dynamics Eq. (13a), yielding

$$\partial_t \varepsilon + \overline{u}\partial_x \varepsilon = -\varepsilon \partial_x \overline{u} + \overline{\varepsilon \partial_x \varepsilon} + \kappa \partial_x^2 \varepsilon - \varepsilon \partial_x \varepsilon. \tag{13b}$$

Hence, the dynamics of the mean flow and of the fluctuations are described by the coupled system Eq. (13).

Note that the term $-\overline{\varepsilon \partial_x \varepsilon}$ is the offset of the mean state due to the fluctuations. The offset term does not affect the statistical properties of the perturbations $\varepsilon$, while it is crucial to the dynamics of $\overline{u}$. From the commutativity of the ensemble mean with spatial derivative, $\overline{\partial_x(\cdot)} = \partial_x \overline{(\cdot)}$, the offset term $\overline{\varepsilon \partial_x \varepsilon} = \frac{1}{2}\partial_x \overline{\varepsilon^2}$ can be written as a function of the variance $V = \overline{\varepsilon^2}$,

$\overline{\varepsilon \partial_x \varepsilon} = \frac{1}{2}\partial_x V$.

If the magnitude of the perturbation $\varepsilon$ is small, Eq. (13) can be simplified into the fluctuation-mean-flow dynamics

$$\partial_t \overline{u} + \overline{u}\partial_x \overline{u} = \kappa \partial_x^2 \overline{u} - \frac{1}{2}\partial_x V, \tag{14a}$$

$$\partial_t \varepsilon + \overline{u}\partial_x \varepsilon = -\varepsilon \partial_x \overline{u} + \frac{1}{2}\partial_x V + \kappa \partial_x^2 \varepsilon, \tag{14b}$$

where the product $\varepsilon \partial_x \varepsilon$ has been discarded while keeping the fluctuation-mean-flow interaction term $\frac{1}{2}\partial_x V = \overline{\varepsilon \partial_x \varepsilon}$. Note that

the tangent-linear dynamics corresponds to Eq. (14) but where the offset term $\frac{1}{2}\partial_x V$ is discarded. Moreover, as pointed out in Ménard (1994), Eq. (14a) is the exact ensemble mean for the Burgers dynamics, while Eq. (14b) is an approximate dynamics. As a consequence, if the variance field is the true one, then the mean predicted by Eq. (14a) is the true ensemble mean (Ménard, 1994, sec. 5.5.2).

The aim is now to determine the dynamics of the two-point error covariance function, $\overline{\varepsilon_x \varepsilon_y}$, which corresponds, after spatial

discretization, to the time evolution of the covariance matrix $\mathbf{P}$ in data assimilation. Following the splitting strategy developed

in Pannekoucke et al. (2016), the evolution of the perturbation $\varepsilon$ is decomposed considering the effect of each process. The splitting strategy is a theoretical method to deduced the so-called infinitesimal generator of an evolution equation, by taking advantage of the Lie-Trotter formula to separate each processes (or appropriate arrangements of the processes). This strategy should not be confused with the numerical time-splitting which introduces numerical errors (Sportisse, 2007). Here, as seen in Eq. (14b), four processes influence the error statistics: (i) a production term due to the transport of the mean flow by the perturbation $-\varepsilon\partial_x\overline{u}$, (ii) the transport of the perturbation by the mean flow $\overline{u}\partial_x\varepsilon$, (iii) a diffusion term $\kappa\partial_x^2\varepsilon$, and (iv) an offset term $\frac{1}{2}\partial_x V$ due to the averaged nonlinear self-interaction of the perturbation $\overline{\varepsilon\partial_x\varepsilon}$.

Since the offset (iv) modifies the mean but not the higher statistical moments of $\varepsilon$ and without loss of generality, only the first three elementary processes are needed for the description of the covariance dynamics:

$$\partial_t\varepsilon = (-\partial_x\overline{u})\varepsilon, \tag{15a}$$

$$\partial_t\varepsilon = -\overline{u}\partial_x\varepsilon, \tag{15b}$$

$$\partial_t\varepsilon = \kappa\partial_x^2\varepsilon. \tag{15c}$$

The effect of each process in Eq. (15) onto the dynamics Eq. (11) of the variance and the local diffusion tensor is now described.

## 3.2 Separate contribution of elementary processes

The contribution of the production term Eq. (15a) is first examined, then the transport Eq. (15b) and finally the diffusion Eq. (15c).

### 3.2.1 Contribution of the production term

The production term describes the amplification of the error due to the gradient of the mean field $\overline{u}$. This process can be viewed as a diagonal operator in the function space where the random field $\varepsilon$ lies. As a consequence, this error dynamics affects the variance but not the metric tensor. This leads to the following parameter dynamics

$$\partial_t V = -(2\partial_x\overline{u})V, \tag{16a}$$

$$\partial_t\nu = 0. \tag{16b}$$

### 3.2.2 Contribution of the transport term

The time evolution of the variance and the diffusion fields due to the transport term Eq. (15b) is now tackled. Since the derivation is archetypal of how to proceed, the calculus are detailed.

The dynamics of error variance fields, deduced from Eq. (11a), yields

$$\partial_t V = 2\overline{\varepsilon(-\overline{u}\partial_x\varepsilon)} = -\overline{u}\partial_x\varepsilon^2. \tag{17}$$

From the commutation of ensemble average and partial derivative, it simplifies to

$$\partial_t V = -\overline{u}\partial_x V. \tag{18}$$

Since $V = \sigma^2$, the dynamics of the standard deviation is given by

$$\partial_t \sigma = -\overline{u}\partial_x \sigma. \tag{19}$$

The dynamics of the metric tensor is deduced from Eq. (11b)

$$
\begin{aligned}
\partial_t g &= \partial_t \overline{(\partial_x \tilde{\varepsilon})^2}, \\
&= 2\overline{\partial_{tx}^2 \tilde{\varepsilon} \, \partial_x \tilde{\varepsilon}} \\
&= 2\overline{\partial_x \left[\partial_t \left(\frac{\varepsilon}{\sigma}\right)\right] \partial_x \tilde{\varepsilon}} \\
&= 2\overline{\partial_x \left[\frac{1}{\sigma}\partial_t \varepsilon - \frac{\varepsilon}{\sigma^2}\partial_t \sigma\right] \partial_x \tilde{\varepsilon}}.
\end{aligned}
$$

With the normalized error $\tilde{\varepsilon} = \frac{1}{\sigma}\varepsilon$ and the dynamics of the standard-deviation Eq. (19), the dynamics of the metric reads

$$
\begin{aligned}
\partial_t g &= 2\overline{\partial_x \left[-\frac{\overline{u}}{\sigma}\partial_x (\sigma\tilde{\varepsilon}) + \tilde{\varepsilon}\overline{u}\partial_x \ln \sigma\right] \partial_x \tilde{\varepsilon}}, \\
&= -2\partial_x \overline{u}\overline{(\partial_x \tilde{\varepsilon})^2} - 2\overline{u}\overline{\partial_x \tilde{\varepsilon}\partial_x^2 \tilde{\varepsilon}}.
\end{aligned}
$$

From the identity $\partial_x(\partial_x \tilde{\varepsilon}\partial_x \tilde{\varepsilon}) = 2\partial_x \tilde{\varepsilon}\partial_x^2 \tilde{\varepsilon}$, and from $g = \overline{(\partial_x \tilde{\varepsilon})^2}$, we obtain

$$\partial_t g + \overline{u}\partial_x g = -2(\partial_x \overline{u})g. \tag{20}$$

Hence, the variance and the local diffusion $\nu = \frac{1}{2g}$ evolve following

$$\partial_t V + \overline{u}\partial_x V = 0, \tag{21a}$$
$$\partial_t \nu + \overline{u}\partial_x \nu = 2(\partial_x \overline{u})\nu. \tag{21b}$$

These equations represent the transport of the variance and of the diffusion by the mean flow: the variance is conserved, while the diffusion tensor is warped by the mean flow.

### 3.2.3 Contribution of the diffusion term

Following the same procedure, the dynamics of $V$ and $g$, Eq. (11), is given for the diffusion process Eq. (15c) by

$$\partial_t V = \kappa\left(\partial_x^2 V - \frac{1}{2V}(\partial_x V)^2\right) - 2g\kappa V, \tag{22a}$$

$$
\begin{aligned}
\partial_t g = {}& 2\kappa g\frac{1}{V}\partial_x^2 V - 2g\kappa\frac{1}{V^2}(\partial_x V)^2 + \kappa\partial_x g\frac{1}{V}\partial_x V + \\
& \kappa\partial_x^2 g + 2g^2\kappa - 2\kappa\overline{(\partial_x^2 \tilde{\varepsilon})^2}.
\end{aligned} \tag{22b}
$$

As it is expected while dealing with Reynold equations, a closure problem appears since the term $\overline{(\partial_x^2\tilde{\varepsilon})^2}$ cannot be deduced from either $V$ or $g$. Hence, a parameterization is needed to pursue.

To proceed further, we take advantage of the link between the unknown quantity $\overline{(\partial_x^2\tilde{\varepsilon})^2}$ and the fourth order term $K_x$ of the Taylor expansion of the error correlation function (see appendix A)

$$\rho(x, x+\delta x) = 1 - \frac{1}{2}g_x\delta x^2 + S_x\delta x^3 + K_x\delta x^4 + o(\delta x^4), \tag{23}$$

where

$$g_x = \overline{\partial_x\tilde{\varepsilon}_x\partial_x\tilde{\varepsilon}_x}, \tag{24a}$$

$$S_x = -\frac{1}{4}\partial_x g_x, \tag{24b}$$

$$K_x = \frac{1}{24}\overline{(\partial_x^2\tilde{\varepsilon}_x)^2} - \frac{1}{12}\partial_x^2 g_x. \tag{24c}$$

The quantities $S_x$ and $K_x$ are later called the skewness and the kurtosis of the correlation function $\rho(x, \cdot)$. Note that, due to the symmetry of the two-points correlation functions, $\rho(x,y) = \rho(y,x)$, the skewness $S_x$ is entirely determined by the metric field $g$. As a result, a choice of the kurtosis implies choosing the closure.

Two particular cases are interesting to discuss: when the random field is statistically homogeneous and, moreover, when the correlation function is a Gaussian function. In the case where the error random field is homogeneous, the error correlation function is homogeneous too: $\rho(x,y) = \rho(x+\delta, y+\delta), \forall\delta\in\mathbb{R}$. As a result, the fields of metric, skewness and kurtosis are constant fields denoted by $g^h$, $S^h$ and $K^h$. Due to the homogeneity of the metric field $g^h$, the skewness Eq. (24b) is zero, and the kurtosis Eq. (24c) is $K^h = \frac{1}{24}\overline{(\partial_x^2\tilde{\varepsilon}_x)^2}$. In the case where the homogeneous correlation function is the Gaussian $\rho_G(x, x+\delta x) = e^{-\frac{\delta x^2}{2L_G^2}}$, where $L_G$ stands for the homogeneous error correlation length-scale, the Taylor expansion reads

$$\rho_G(x, x+\delta x) = 1 - \frac{1}{2L_G^2}\delta x^2 + \frac{1}{2}\left(\frac{1}{2L_G^2}\right)^2\delta x^4 + o(\delta x^4), \tag{25}$$

and by identification with Eq. (23), $g_G = \frac{1}{L_G^2}$, $S_G = 0$ and $K_G = \frac{1}{8}g_G^2$.

We propose to use these results to formulate a closure model: for a general smooth error random field of metric field $g_x$, the kurtosis $K_x$, Eq. (24c), is approximated by

$$K_x^{GC} = \frac{1}{8}g_x^2 - \frac{1}{12}\partial_x^2 g_x, \tag{26}$$

where the first term of the right hand side is the kurtosis of the equivalent local homogeneous Gaussian correlation function. This closure is hereby called the *locally homogeneous Gaussian closure* or simply the *Gaussian closure*.

With this Gaussian closure Eq. (26), the dynamics of diffusion Eq. (22) is

$$\partial_t V = \kappa\left(\partial_x^2 V - \frac{1}{2}\frac{1}{V}(\partial_x V)^2\right) - 2g\kappa V, \tag{27a}$$

$$\partial_t g = 2\kappa g\frac{1}{V}\partial_x^2 V - 2\kappa g\frac{1}{V^2}(\partial_x V)^2 + $$
$$\kappa\partial_x g\frac{1}{V}\partial_x V + \kappa\partial_x^2 g - 4g^2\kappa. \tag{27b}$$

In the one dimensional case, the dynamics of the local diffusion tensor are deduced from the dynamics of the metric from $\partial_t \nu = -2\nu^2 \partial_t g$ with $g = \frac{1}{2\nu}$. Thus, the variance and local diffusion tensor evolution are equivalently expressed as

$$\partial_t V = \kappa \left( \partial_x^2 V - \frac{1}{2V}(\partial_x V)^2 \right) - \frac{\kappa}{\nu} V, \tag{28a}$$

$$\partial_t \nu = 2\kappa - 2\frac{\kappa\nu}{V}\partial_x^2 V + 2\frac{\kappa\nu}{V^2}(\partial_x V)^2 + $$

$$\kappa\frac{1}{V}\partial_x V \partial_x \nu + \kappa\partial_x^2 \nu - 2\kappa\frac{1}{\nu}(\partial_x \nu)^2. \tag{28b}$$

Contrary to the production Eq. (16) and the transport Eq. (21) processes, the effect of the diffusion process Eq. (28) creates a nonlinear coupling between the variance and the local diffusion field.

The parametric covariance dynamics for the Burgers equation is now expressed collecting all these results.

### 3.3 Parametric covariance dynamics of the Burgers equation

From Eq. (16), Eq. (21), and Eq. (28), the complete parametric covariance dynamics for the Burgers equation under the Gaussian closure, is given by the coupled system

$$\partial_t \overline{u} + \overline{u}\partial_x \overline{u} = \kappa\partial_x^2 \overline{u} - \frac{1}{2}\partial_x V, \tag{29a}$$

$$\partial_t V + \overline{u}\partial_x V = -2(\partial_x \overline{u})V + \kappa\partial_x^2 V - \frac{\kappa}{2}\frac{1}{V}(\partial_x V)^2 - \frac{\kappa}{\nu}V, \tag{29b}$$

$$\partial_t \nu + \overline{u}\partial_x \nu = 2(\partial_x \overline{u})\nu + 2\kappa - 2\frac{\kappa\nu}{V}\partial_x^2 V + 2\frac{\kappa\nu}{V^2}(\partial_x V)^2$$

$$+ \kappa\frac{1}{V}\partial_x V \partial_x \nu + \kappa\partial_x^2 \nu - 2\kappa\frac{1}{\nu}(\partial_x \nu)^2. \tag{29c}$$

Equation (29) exhibits a nonlinear coupling between the variance, Eq. (29b), and the local diffusion tensor, Eq. (29c), which illustrates the intricacy of the action of the diffusion process on the error dynamics. Moreover, Eq. (29) differs from its tangent-linear equivalent by the term $-\frac{1}{2}\partial_x V$ in Eq. (29a).

A numerical experiment is now proposed to illustrate and assess these theoretical results.

## 4 Numerical experiment

A numerical experiment is proposed to illustrate the ability of the PKF forecast to reproduce the statistical evolution of the errors in the diffusive Burgers model. The numerical setting is first introduced, followed by an evaluation of the kurtosis closure. Then, the PKF is assessed using a large ensemble of nonlinear forecasts (6400 members). A sensitivity test on the different terms in the PKF concludes the section.

### 4.1 Numerical setting

For the numerical validation, a front-like situation is considered on a periodic domain of length $D = 1000$ km, discretized with $N = 241$ grid-points. The initial reference state, shown in Fig. 1, is the velocity field $\overline{u^0}(x) = U_{\max}\left[1 + \cos(2\pi(x - D/4)/D)\right]/2$

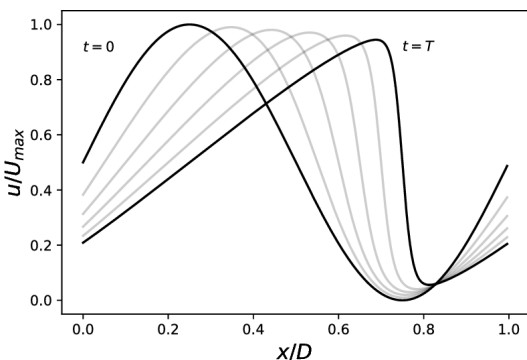

**Figure 1.** Nonlinear solution of the diffusive Burger' equation for the times $t = \{0, 0.2, 0.4, 0.6, 0.8, 1\}T$.

with $U_{\max} = 20$ km/h. From the nonlinear forecast of Eq. (12) starting from $\overline{u^0}$, the maximum initially at 250 km, develops a front structure at 750 km after $T = 24$ hours of forecast. The simulations considered here are integrated from the initial time $t = 0$ to the final time $t = T$. The Burgers equation Eq. (12) has been numerically integrating considering a classical finite difference spatial scheme and a fourth order Runge-Kutta time scheme with $\delta t = 0.002T$ and $\kappa = 0.0025 D^2/T$.

The random perturbation at initial time, $\varepsilon^0$, is set as a homogeneous random field of Gaussian distribution. Following Gaspari and Cohn (1999), the homogeneous correlation function is set, in accordance with the geometry of the circle, as the chordal distance Gaussian correlation

$$\rho^0(x,y) = e^{-\frac{d(x,y)^2}{2L_{\mathrm{G}}^2}},\tag{30}$$

where the homogeneous correlation length-scale is $L_{\mathrm{G}} = 20$ km and where $d(x,y) = \frac{D}{\pi}\left|\sin\frac{\pi}{D}(x-y)\right|$ is the chordal distance
between the two geographical positions $x$ and $y$. Since the length-scale $L_{\mathrm{G}}$ is much smaller than the perimeter $D$, the Gaussian correlation Eq. (30) with the arc-length distance $d(x,y) = |x-y|^2$ is numerically very closed to the one with the chordal distance (while at the theoretical level, the arc-length Gaussian is not strictly a correlation function on the circle, see Gaspari and Cohn (1999)), and leads to the same numerical results.

The covariance function is then defined as

$$P^0(x,y) = (\sigma^0)^2 \rho^0(x,y),\tag{31}$$

where $\sigma^0$ is the constant standard-deviation field. Thereafter, four magnitudes of standard deviation $\sigma^0$ are considered: $\sigma^0_{1\%} = 0.01\,U_{\max}$, $\sigma^0_{10\%} = 0.1\,U_{\max}$, $\sigma^0_{20\%} = 0.2\,U_{\max}$, and $\sigma^0_{50\%} = 0.5\,U_{\max}$.

The time evolution of the true error covariance functions is computed considering a large ensemble of $N_e$ nonlinear forecasts of Eq. (12) with $N_e = 6400$. From the non-parametric convergence, the expected sampling error should thus represents about
$1/\sqrt{N_e} = 1.25\%$ of the real statistics. In order to limit the differences when comparing the results and due to the sampling noise, a single large ensemble of normalized error $(\tilde{\varepsilon}_k)_{k \in [1, N_e]}$ has been generated as $\tilde{\varepsilon}_k = \mathbf{C}^{1/2}\zeta_k$ where $\mathbf{C}^{1/2}$ is the square-root of the correlation matrix deduced from the correlations Eq. (30), and $\zeta_k$ is a sample of a random Gaussian noise of zero

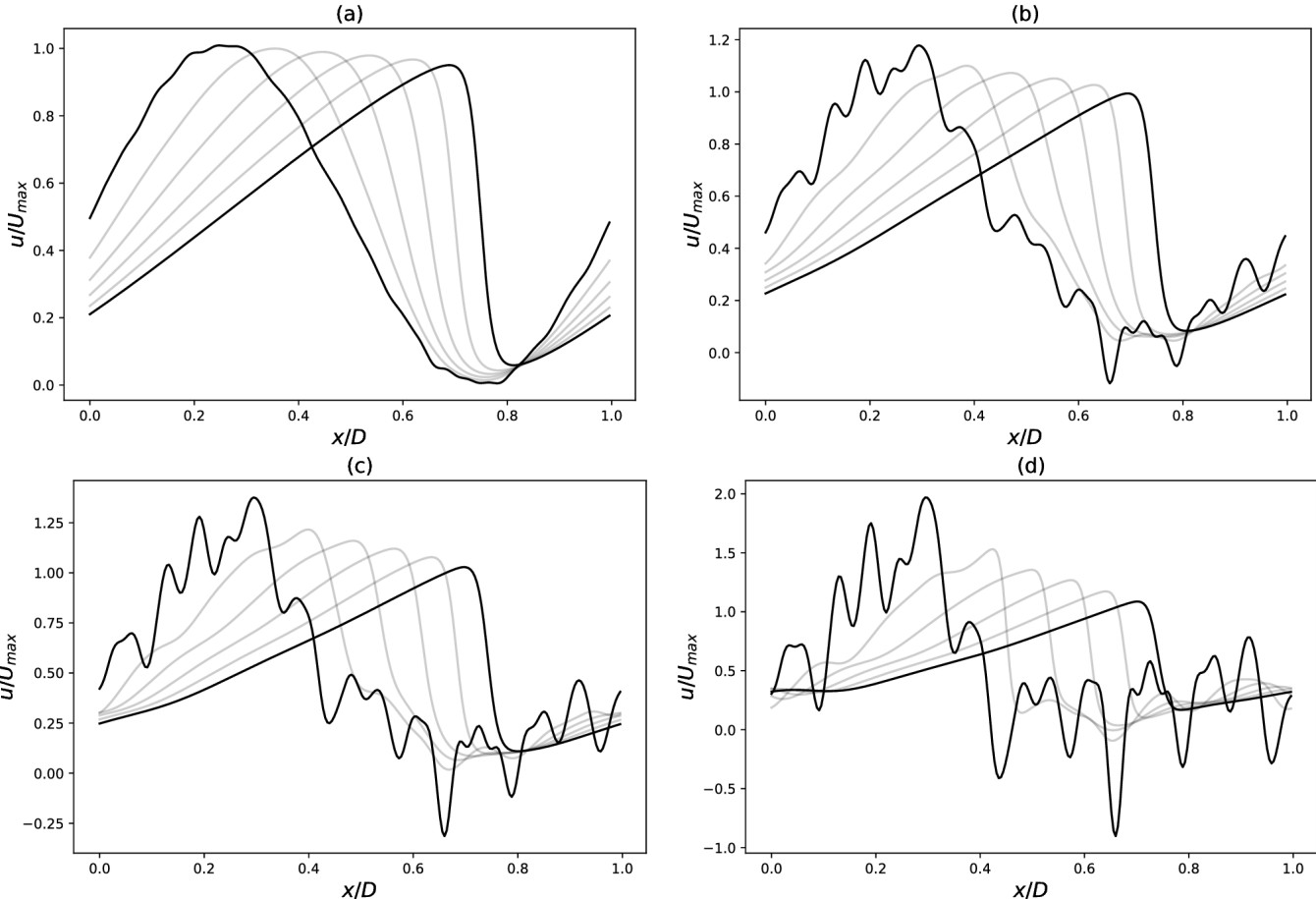

**Figure 2.** Solutions from the initial perturbations of magnitude (a) $\sigma^0_{1\%}$, (b) $\sigma^0_{10\%}$, (c) $\sigma^0_{20\%}$ and (d) $\sigma^0_{50\%}$, for the times $t = \{0, 0.2, 0.4, 0.6, 0.8, 1\}T$.

mean and covariance matrix the identity matrix. The ensemble of initial perturbation is then generated as $\varepsilon_k = \sigma^0 \tilde{\varepsilon}_k$ with $\sigma^0 \in \{\sigma^0_{1\%}, \sigma^0_{10\%}, \sigma^0_{20\%}, \sigma^0_{50\%}\}$.

Since the parametric covariance dynamics Eq. (29) has been theoretically derived for small perturbations, it has to be compared with the statistics from the ensemble of small magnitude noise. Hence the validation is later conducted by considering the ensemble generated from the initial standard deviation $\sigma^0_{1\%}$. Limits of predictability of the parametric covariance dynamics Eq. (29) are also addressed considering the ensembles of larger initial uncertainty, from $\sigma^0_{10\%}$ to $\sigma^0_{50\%}$.

Figure 2 illustrates the time evolution of the four perturbed initial conditions whose perturbations are generated from the normalized pertubation $\tilde{\varepsilon}_1$ scaled with the initial standard deviations from $\sigma^0_{1\%}$ in panel (a) to $\sigma^0_{50\%}$ in panel (d).

These ensembles are first used to tackle the closure of kurtosis, as discussed now.

## 4.2 Evaluation of the kurtosis closure

The aim of this section is to compare the kurtosis diagnosed from the true error covariance Eq. (24c), with the kurtosis resulting from the Gaussian closure Eq. (26). This validation is a crucial step since the quality of the closure will affect the skill of the parametric covariance dynamics Eq. (29). Even though the closure is likely to be wrong for an arbitrary covariance matrix,
it is expected to apply to most statistics encountered in applications. The large ensemble, whose the initial perturbations are sampled by using $\sigma^0 = \sigma^0_{10\%}$, is considered for the validation. The results are equivalent for the other error magnitudes. For this experiment, the error covariance matrices at times $t = 0$ and $t = T$ are representative of intermediate covariance matrices ; the computation of the true kurtosis and its closure is achieved by considering the ensemble at both times. The computation of the closure Eq. (26) relies on the local metric tensor, which has to be diagnosed from the ensemble.
The local metric and kurtosis can be computed from the ensemble considering Eqs. (24a) and (24c). It is also possible to compute these quantities from the direct estimation of the local correlation function expansion Eq. (23), with the benefit of validating the theoretical derivations made in Eq. (24b) for the skewness and Eq. (24c) for the kurtosis. This motivates the estimation of these quantities from the computation of local polynomial expansions, which are computed as follows.

For each position $x$, the fourth order polynomial approximation of the correlation function $\rho(x, \cdot)$ is estimated as the La-
grange interpolating polynomial $Q_x(X) = \sum_{k=-2}^{2} \rho(x, x_k) \Pi_{i \neq k} \frac{(X - x_i)}{(x_k - x_i)}$, computed from the five correlation values $(\rho(x, x_k))_{k \in [-2, 2]}$ where $x_k = x + k\delta x$. Expanding the Lagrange interpolating polynomial as $Q_x(X) = p_0 + p_1 X + p_2 X^2 + p_3 X^3 + p_4 X^4$, one obtains $g_x = -2p_2$ and $K_x = p_4$.

Figure 3 illustrates the results computed from the ensemble at $t = 0$ (panel a) and $t = T$ (panel b), showing in continuous line the length-scale $L_x = 1/\sqrt{g_x}$ (top panels) and the kurtosis (bottom panels), normalized by the initial homogeneous Gaussian
values $L_G$ and $K_G = g_G^2/8$ with $g_G = 1/L_G^2$. The kurtosis' closure Eq. (26) computed from the metric is in dashed line (bottom panels).

At $t = 0$, the length-scale (respectively the kurtosis) field is homogeneously equal to the initial values $L_G$ (respectively $K_G$). The small fluctuations visible at this time are due to the sampling noise. For $t = T$, the length-scale is larger than at the initial time, and presents an area of small values in the vicinity of the front position $x = 0.75D$. The kurtosis is negligible, except at
the vicinity of the front position, where the field is oscillating. For both times, it appears that the kurtosis' closure is able to reproduce the main behavior of the true kurtosis, with a low relative error $||K - K^{GC}||/||K||$ of 5.2% (respectively 21.5%) at time 0 (respectively $T$), where $|| \cdot ||$ stands for the $L^2$ norm. At time $T$, the maximum difference between the two normalized kurtosis, is 0.05.

Note that all the previous results are similar for the smaller initial uncertainty magnitude $\sigma^0_{1\%}$, with a relative error of 5.2%
(respectively 14.5%) at time 0 (respectively $T$) (not shown here). For the larger error magnitudes $\sigma^0_{20\%}$ and $\sigma^0_{50\%}$, the relative error at time $T$ is respectively 40% and 63%. Hence, for this numerical simulation, the Gaussian closure proposed for the kurtosis appears relevant to approximate the real feature of the correlation shape. This is now used to explore the ability of the PKF to accurately predict the error statistics.

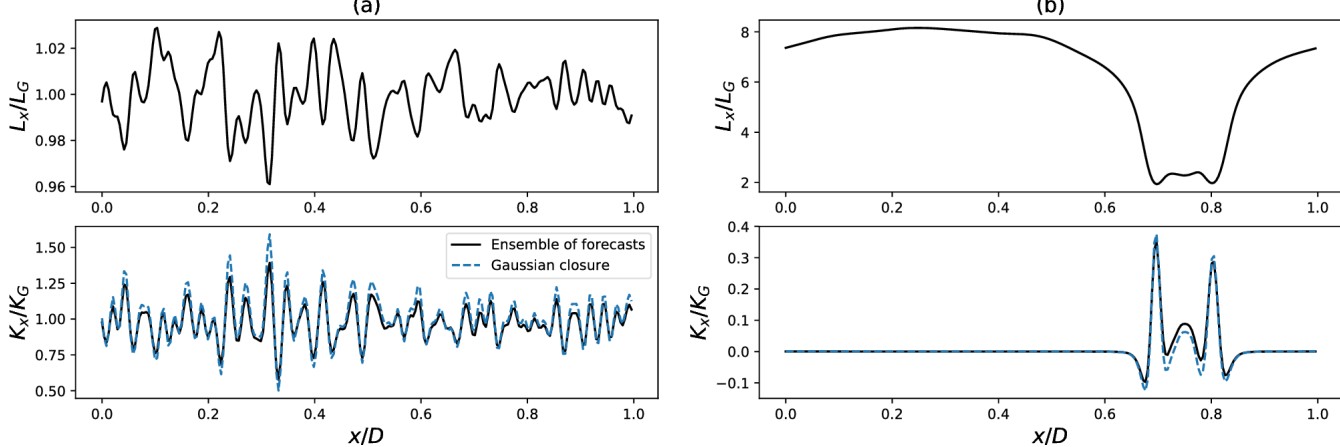

**Figure 3.** Diagnoses of the length-scale $L_x$ normalized by $L_G$ (top panels) and of the kurtosis normalized by $K_G = \frac{1}{8} g_G^2$ (bottom panels) deduced from the ensemble of forecast with initial uncertainty magnitude $\sigma_{10\%}^0$, at times $t = 0$ (a) and $t = T$ (b) (continuous line). The kurtosis is compared with its closure $K_x^{GC} = \frac{1}{8} g_x^2 - \frac{1}{12} \partial_x^2 g_x$ (dashed line) computed from the metric field.

## 4.3 Parametric versus ensemble statistics

The parametric setting is based on the time integration of the nonlinear coupled system Eq. (29) considering an equivalent numerical scheme than the one solving the Burgers equation Eq. (12), *i.e.* finite difference and RK4, with the same time step as detailed in Section 4.1. The numerical cost is of the order of a nonlinear time integration of the nonlinear Burgers equation. In this one-dimensional case, only two scalar fields are propagated: the variance $V$ and the local diffusion field $\nu$.

The mean, the error variance and length-scale fields are reproduced in Fig. 4, Fig. 5 and Fig. 6, respectively, considering a range of initial errors. These figures compare the diagnosis from the ensemble of nonlinear forecasts of Eq. (12) with the statistics predicted by the parametric model Eq. (29). The mean diagnosed from the ensemble and predicted by the parametric model is first considered.

### 4.3.1 Comparison of the means

In order to appreciate the differences between the ensemble and the parametric means, the discussion is focused on the results at the final time $T$. When the initial error magnitude is small (Fig.4-(a) ), corresponding to the tangent linear regime, the ensemble mean (continuous line) and the mean state predicted from the parametric model (dashed line) coincide with the reference state $u^0(T, x)$ (grey solid line, reproduced from Fig. 1). This is within the tangent linear validity regime where the small magnitude of the fluctuation has not impact on the ensemble mean which is then equal to the reference trajectory. For larger error magnitudes, the ensemble mean is expected to deviate from the reference trajectory due to the nonlinear interaction between the fluctuation and the mean. In the Burgers equation, the deviation is due to $-\frac{1}{2} \partial_x V$ in Eq. (29a) which implies here that the ensemble mean decreases as if the diffusion increased with the error magnitude (panels (b-d) ). The mean predicted by

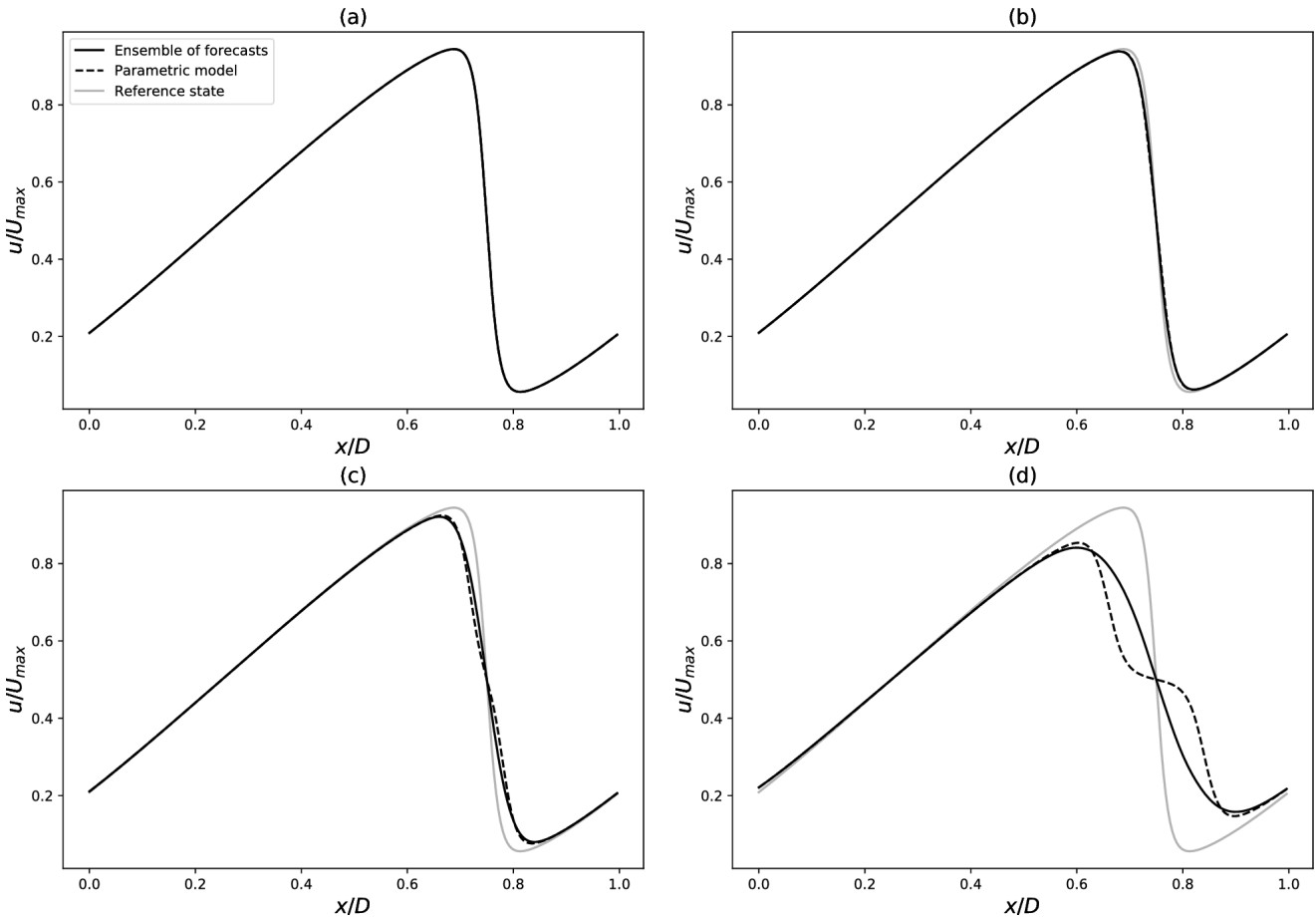

**Figure 4.** Mean state at time $t = T$ computed from the ensemble of forecasts (solid line) and predicted from the parametric model Eq. (29) (dashed line), for the initial perturbations of magnitude (a) $\sigma^0_{1\%}$, (b) $\sigma^0_{10\%}$, (c) $\sigma^0_{20\%}$ and (d) $\sigma^0_{50\%}$. The reference state is in grey solid line.

the parametric model is very close to the ensemble mean (panel (b-c) ) for the moderate error magnitudes of $\sigma_{10\%}$ and $\sigma_{20\%}$, but presents an anomalous distortion at the inflexion point for the larger error magnitudes (panel (d) ). Hence, for the particular case of the Burgers dynamics, the parametric prediction of the mean is an accurate approximation of the ensemble mean, even for mild error magnitudes.

5    ### 4.3.2    Comparison of the variance and length-scale statistics

The variance (Fig. 5) and length-scale (Fig. 6) statistics are now discussed. For the small error magnitude, as seen in panels (a), the uncertainty is spreading at the initial time due to the physical diffusion, resulting in a strong dampening of the variance. This is accompanied by a global increase of the length-scale, except in the vicinity of the inflection point located near $x = 0.5D$ (see Fig. 1). Then, as time goes on, the dynamical front generates a source of uncertainty where a beam of variance appears

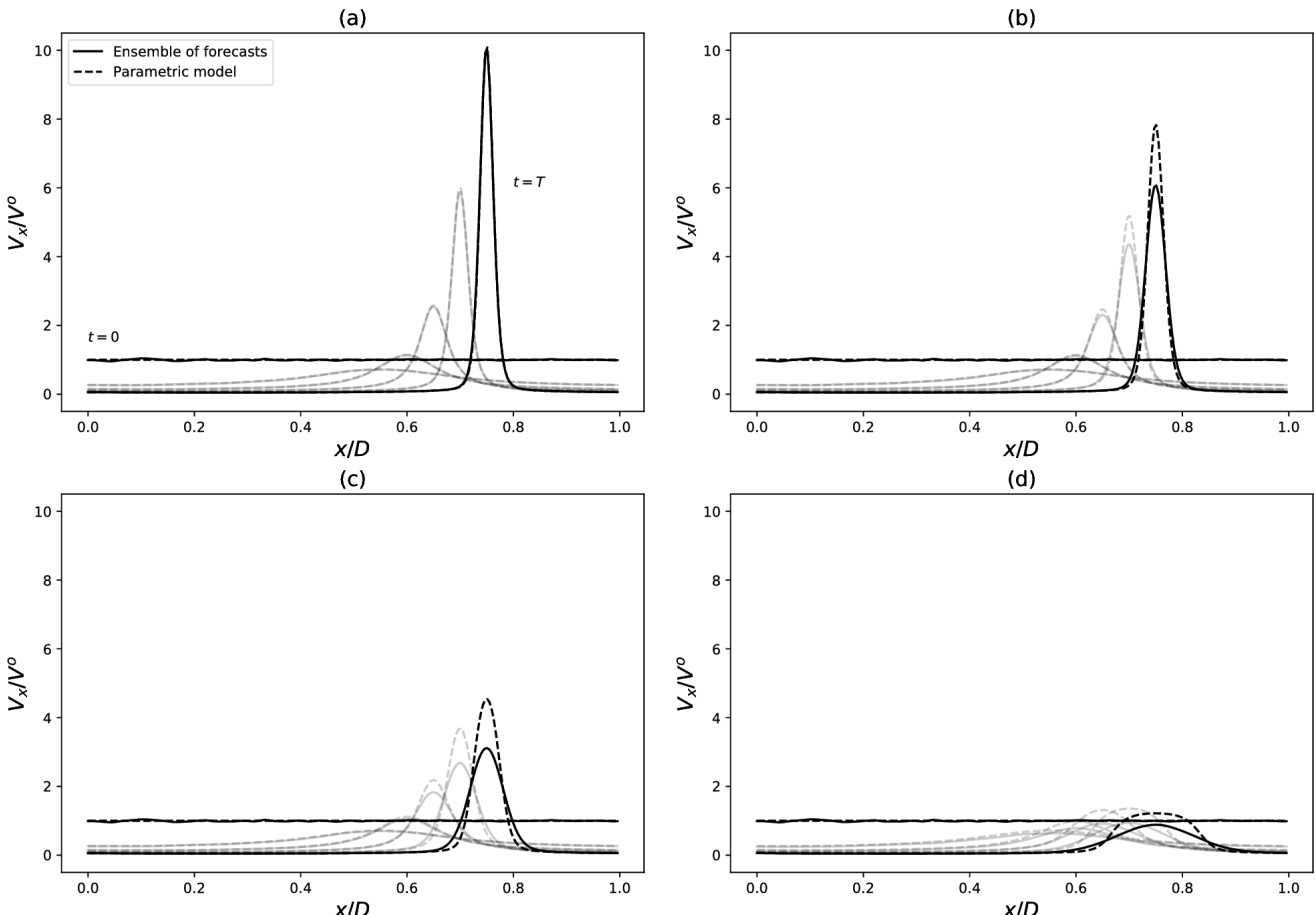

**Figure 5.** Parametric (dashed line) versus ensemble estimated (continuous line) variance fields for initial perturbations of standard deviation magnitude $\sigma_{1\%}^0$ (a) (tangent-linear dyanmics), $\sigma_{10\%}^0$ (b) (weakly nonlinear dynamics), $\sigma_{20\%}^0$ (c) and $\sigma_{50\%}^0$ (d). The fields are represented only for the times $t = \{0, 0.2, 0.4, 0.6, 0.8, 1\}T$.

and increases with time, yielding a maximum of $10.0$ times the initial variance at $t = T$. The length-scale remains short close to the front position, except for a peak emerging from time $t = 0.6T$, evolving with the flow at the inflection point. Comparing to the ensemble statistics, the PKF is able to capture all the details of the dynamics. This strongly support Eq. (29) as well as the underlying assumptions: the derivation of the tangent-linear dynamics for the error variance and length-scale fields, and the

5   Gaussian closure for the kurtosis. In particular, the area of large length-scale values visible at the inflection point of the front, is a real signal and not a numerical artifact of the diagnosis, since it is produced in both simulations.

The case of the error magnitude of $\sigma_{10\%}$, where the tangent-linear approximation should no more be valid, is now considered (see panels (b) ). Key features previously described are still present: emergence of a beam of uncertainty, increase of the length-scale except in the vicinity of the front. However, two differences appear compared to the ensemble statistics reference. Firstly,

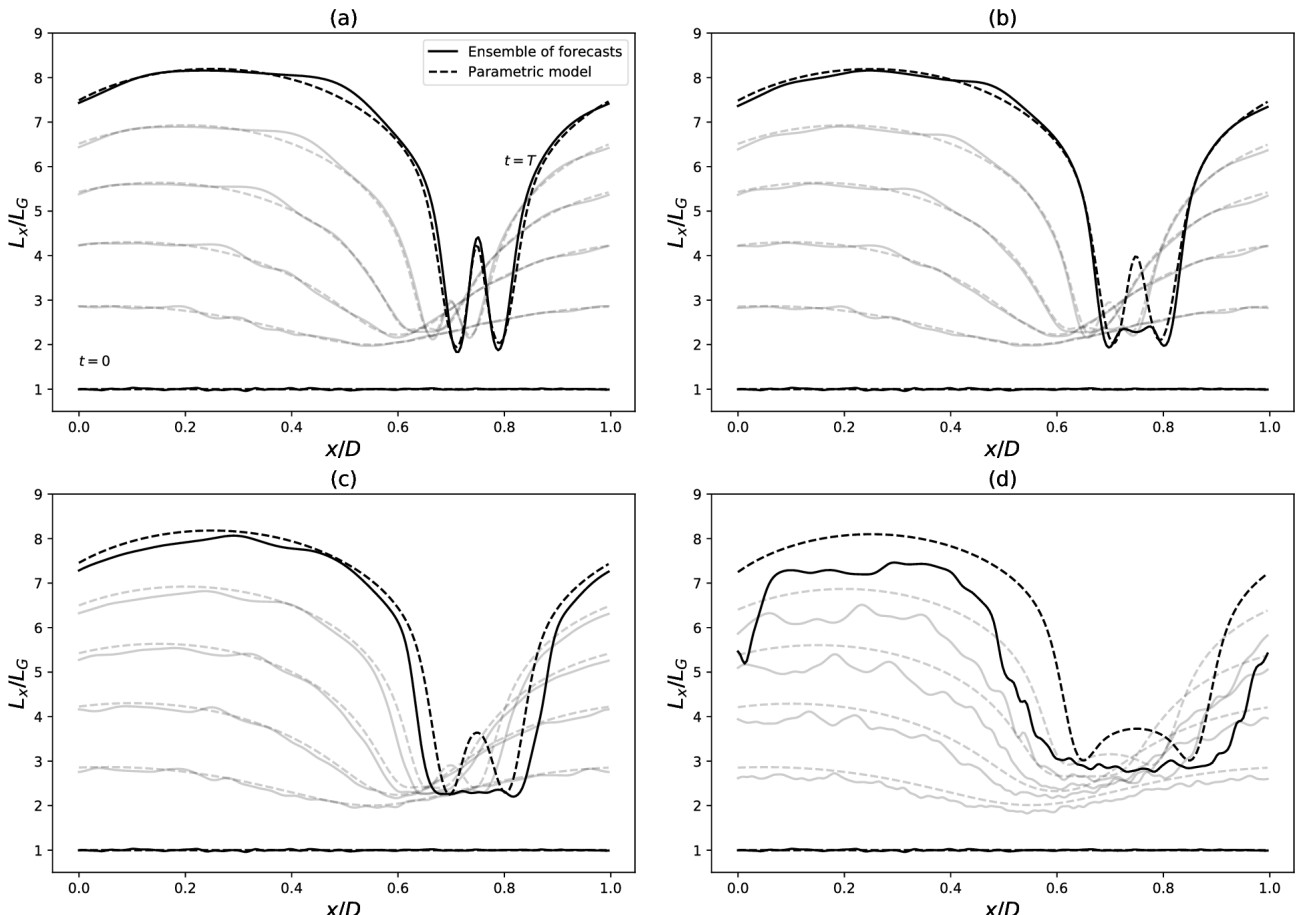

**Figure 6.** Parametric (dashed line) versus ensemble estimated (continuous line) length-scale field for initial standard deviations $\sigma^0_{1\%}$ (a) (tangent-linear dynamics), $\sigma^0_{10\%}$ (b) (weakly nonlinear dynamics), $\sigma^0_{20\%}$ (c) and $\sigma^0_{50\%}$ (d). The fields are represented only for the times $t = \{0, 0.2, 0.4, 0.6, 0.8, 1\}T$.

the magnitude of the uncertainty is lower than in the tangent-linear case; the maximum of variance beam at $t = T$ is now close to $6.0$ times the initial variance. Secondly, the local large length-scale value as depicted in the tangent-linear setting is nearly flat at the bottom of the small length-scale basin associated to the front. The main features of the PKF predictions are recovered: the variance beam has lower magnitude than in the tangent-linear case, and there is still a low length-scale area near the front.

5    Beyond the variance attenuation, a maximum at $t = T$ of $7.8$ times the initial variance, is much greater than the ensemble statistics result, with a relative error of $29\%$. Moreover, the length-scale field displays a peak at the front, similar to the one described for the tangent-linear.

In order to assess the role of the nonlinear term, $\varepsilon\partial_x\varepsilon$, in the error dynamics Eq. (13b), an evaluation with an ensemble has been performed. The results are displayed in Fig. 7, which shows, at $t = T$, the error variance and length-scale fields estimated

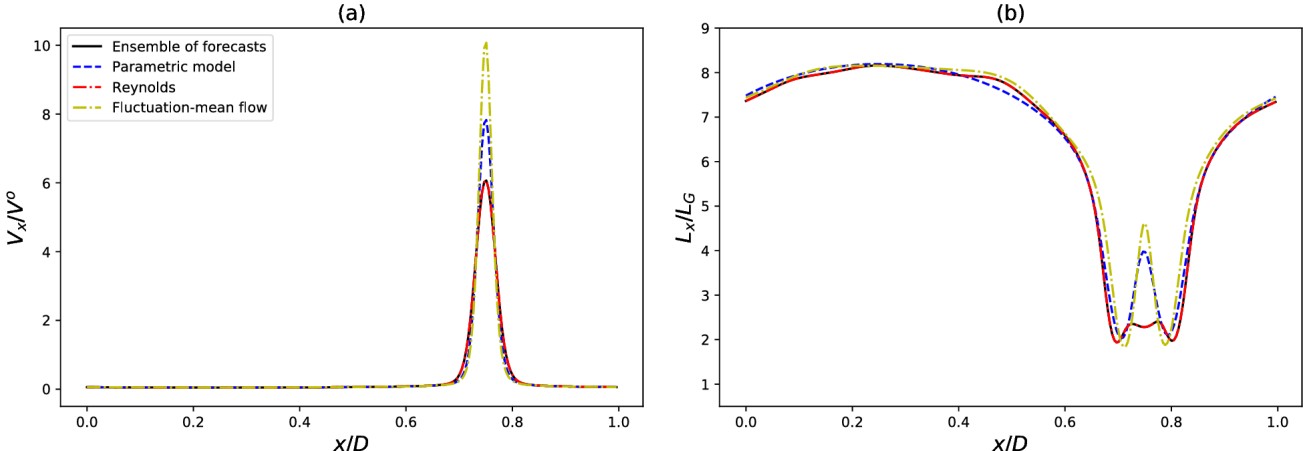

**Figure 7.** Verification of the parametric variance (a) and correlation length-scale (b) at $t = T$ and for an initial perturbation standard deviation $\sigma^0_{10\%}$. The statistics from the ensemble of nonlinear forecasts of Eq. (12) are in black line, the statistics predicted by the parametric model are in blue dashed line, the statistics from the Reynolds equations Eq. (13) (respectively from the fluctuation-mean-flow dynamics Eq. (14)) with (respectively without) the second order term $\varepsilon \partial_x \varepsilon$ is in small red dashed-dotted line (respectively yellow dash-dotted line).

with Eq. (29) (dashed line), compared to the fields diagnosed from a large ensemble (continuous line, and also shown in Fig. 5-(b) and 6-(b)). Then it shows the statistics computed from an ensemble of forecast of the tangent-linear dynamics Eq. (14) (small dashed line) (finite difference and RK4), and using the Reynolds equations Eq. (13) (dash-dotted line) (also finite difference and RK4). It appears that the statistics computed from the tangent-linear dynamics are equivalent to the error

variance and length-scale fields predicted by the parametric model, while the statistics from the Reynolds equations equal those deduced from the ensemble of nonlinear forecasts of Eq. (12). Hence, the difference is well explained by the contribution of the nonlinear term $\varepsilon \partial_x \varepsilon$.

The case of the larger initial error magnitudes of $\sigma_{20\%}$ and $\sigma_{50\%}$ show similar results to the magnitude $\sigma_{10\%}$ case: the small length-scale area is captured by the PKF but with a spurious oscillation not present in the ensemble estimation (Fig. 6, panels

c-d), the position of the beam of uncertainty is well predicted by the PKF but with a larger magnitude than the ensemble estimation (Fig. 5, panels c-d). Since the magnitude of the variance predicted by the PKF seems to increase faster than the ensemble estimation, it is interesting to investigate what happens for longer time window.

### 4.3.3 Long term behaviour

The increase of the PKF variance prediction might be a side effect due to the tangent linear like derivation of the PKF which

could fails to predict the saturation of the error magnitude. In order to tackle the long term behaviour a comparison is conducted with a longer time window of $[0, 5T]$. Since the location of the uncertainty beam is well predicted by the PKF, the comparison focuses on the magnitude of the variance fields maximum. The time series of the variance maximum predicted by the PKF and estimated from the ensemble is shown in Fig. 8. The time evolution is equivalent for all the initial error magnitudes: after a

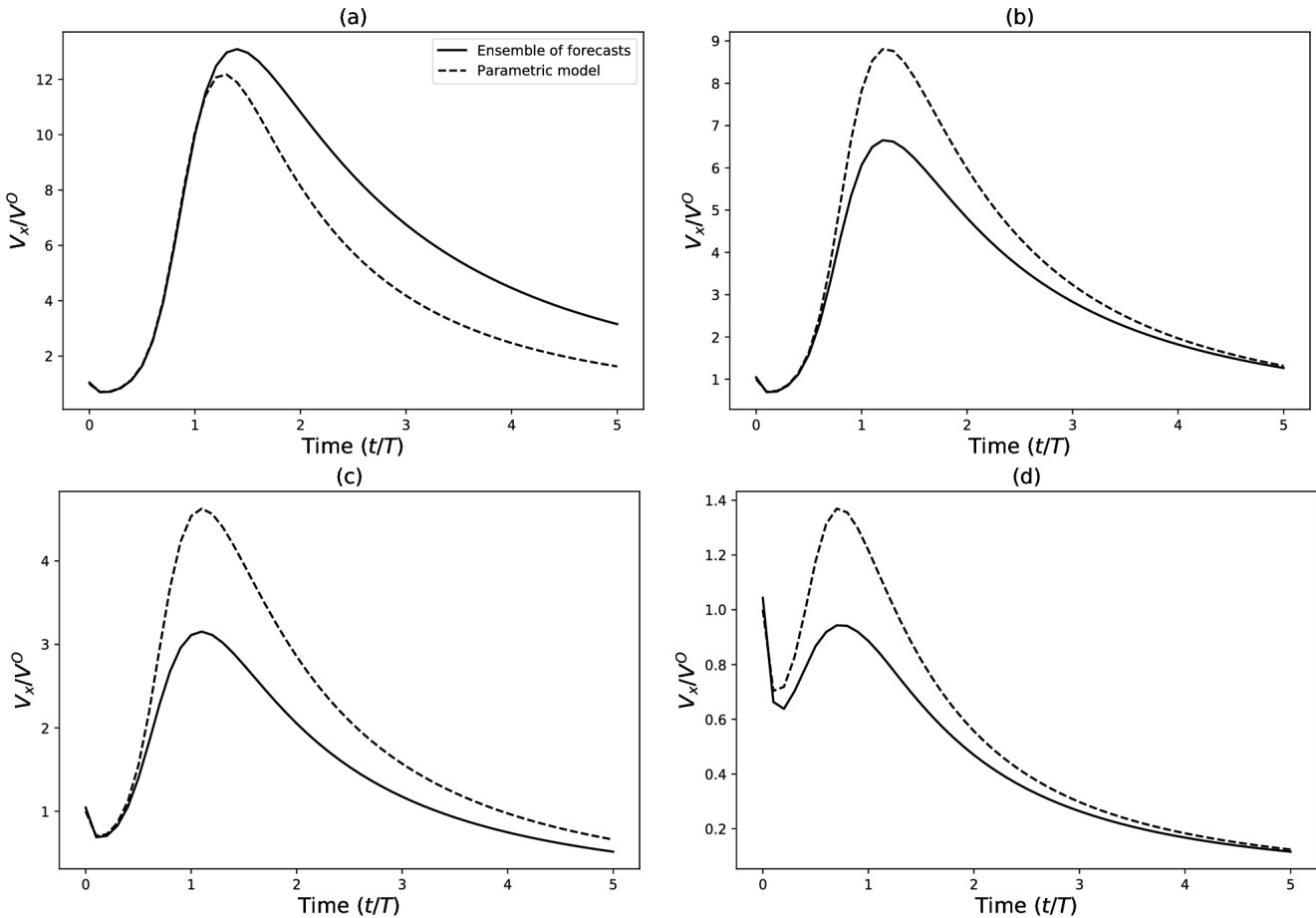

**Figure 8.** Paramatric (dashed line) versus numerical (continuous line) time series of the variance fields' maximum, from $t = 0$ to $t = 5T$, for initial perturbation of standard deviation magnitude $\sigma^0_{1\%}$ (a), $\sigma^0_{10\%}$ (b), $\sigma^0_{20\%}$ (c) and $\sigma^0_{50\%}$ (d).

short transition (where the variance decreases), two phases are seen, where the variance increases (phase 1), then saturates and decreases in long term (phase 2). The time where the maximum of variance is reached shifts with the magnitude of the initial error: it occurs after (respectively before) the time $T$ for $\sigma_{1\%}$ (respectively $\sigma_{50\%}$). We associate the increase to the advection contribution that involves the source term $-2(\partial_x \overline{u})V$ in the variance dynamics Eq. (29b), while the damping is related to the

5   diffusion. Thanks to the competition between the advection and the diffusion, the increase of the variance first saturates then decreases. The PKF reproduces the two phases with a magnitude prediction closed but different from the ensemble estimation. For the small initial error magnitude $\sigma_{1\%}$ the PKF underestimates the variance for the long term behaviour, while the variance is overestimated for larger initial error magnitudes. From numerical investigations with smaller error magnitude at the initial time, the PKF prediction of the phase 2 appears more difficult than for the phase 1. This could be related to the choice of

10   closure made for the kurtosis: while the locally homogeneous Gaussian closure is in accordance with the one diagnosed from

the ensemble, an heterogeneous closure might improve the results. Beyond these deficiencies, it is interesting to retain that the theoretical derivation of the parametric model, which is partly based on the tangent linear assumption, is able to capture the main part of the uncertainty dynamics in the Burgers equation.

### 4.3.4 Discussion

From these results, we can conclude that the PKF forecast, as implemented by Eq. (29), reproduces the tangent-linear evolution of the statistics as given by the covariance forecast equation Eq. (3). Put differently, the PKF forecast reproduces the tangent-linear covariance dynamics occurring in the extended Kalman filter. Since the PKF forecast model is deduced from a small error expansion, it is not meant to recover strongly nonlinear effects, which has been verified in the numerical experiment. Yet, even when the nonlinearity is stronger and the tangent-linear assumption is invalid, the solution of the PKF still shares some

features with the empirical ensemble statistics. This may not be the case anymore for long-time integration of more complex geophysical dynamics. However, this suggests that some of the statistics could be predicted, at least for medium range forecast.

## 5 Conclusions

This study focused on the forecast step of the parametric Kalman filter (PKF) applied to the nonlinear dynamics of the diffusive Burgers equation. The parametric approach consists in approximating the error covariance matrix by a covariance model with

evolving parameter fields. Here the covariance model considered is based on the diffusion equation, parameterized by the error variance and local diffusion fields. Hence, the forecast of the error covariance matrix, which is computationally very demanding in real application with high-dimensional systems, amounts to the forecast of the error variance and local-diffusion fields, whose numerical cost is of the order of a single nonlinear forecast. In comparison, ensemble methods need dozens of members for the covariance forecast (which could be parallelized though), as well as localization to address the rank deficiency.

The derivation of the PKF dynamics has first been rigorously deduced from the dynamics of the perturbation under a small error magnitude assumption. However, a closure problem appears due to the physical diffusion process. This closure issue has been related to the fourth term in the Taylor expansion of correlation function, the kurtosis, and a closure has been proposed based on an homogeneous Gaussian approximation for the kurtosis.

Numerical experiments where the true covariance evolution has been diagnosed from an ensemble forecast have been per-

formed. A comparison with the PKF prediction has first shown the relevance of the closure, even for large error magnitudes. Moreover, these experiments have demonstrated the ability of the parametric formulation to reproduce the main features of the error dynamics when the tangent-linear approximation is valid. When the tangent-linear dynamics is no more valid, the PKF can only reproduce a part of the error statistics evolution, at least for mid-term forecast.

This contribution is a step toward the PKF formulation of more complex dynamics in geophysics. From the present study,

we learned the difficulties of handling the higher order derivatives, since the coupling between the error variance and diffusion fields has been due to the physical diffusion. The Gaussian closure, similar to the one introduced in the kurtosis' treatment, will be useful to provide prognostic dynamics. But we expect that the main difficulties will be encountered in the forecast of multi-

variate statistics that govern the balance between geophysical fields. Theoretically, the PKF formulation enables the forecast of covariance matrices in high dimension. Hence, it might offer new theoretical tools to approximate and to investigate important aspects of the dynamics of errors, such as the unstable subspace of chaotic dynamics. These points will be investigated in further developments.

## 5    Appendix A:  Correlation fourth order derivative

The aim of this section is to show the following theorem (given here for d=1).

**Theorem:** *For smooth and centered error random field $\varepsilon$, with a smooth normalized error random field defined by $\tilde{\varepsilon}_x = \varepsilon_x/\sigma_x$, the error correlation function $\rho(x,y) = \overline{\tilde{\varepsilon}_x \tilde{\varepsilon}_y}$ locally expands as*

$$\rho(x, x + \delta x) = 1 - \frac{1}{2}g_x \delta x^2 + S_x \delta x^3 + K_x \delta x^4 + o(\delta x^4), \tag{A1}$$

*with*

$$g_x = \overline{\partial_x \tilde{\varepsilon}_x \partial_x \tilde{\varepsilon}_x}, \tag{A2a}$$

$$S_x = -\frac{1}{4}\partial_x g_x, \tag{A2b}$$

$$K_x = \frac{1}{24}\overline{\partial_x^2 \tilde{\varepsilon}_x \partial_x^2 \tilde{\varepsilon}_x} - \frac{1}{12}\partial_x^2 g_x. \tag{A2c}$$

Proof: For smooth normalized error field, the Taylor expansion, reads

$$\tilde{\varepsilon}_{x+\delta x} = \tilde{\varepsilon}_x + \partial_x \tilde{\varepsilon}_x \delta x + \frac{1}{2}\partial_x^2 \tilde{\varepsilon}_x \delta x^2 + \frac{1}{6}\partial_x^3 \tilde{\varepsilon}_x \delta x^3 +$$

$$\frac{1}{24}\partial_x^4 \tilde{\varepsilon}_x \delta x^4 + o(||\delta x||^4). \tag{A3}$$

Multiplication by $\tilde{\varepsilon}_x$ yields the correlation function

$$\rho(x, x + \delta x) = \overline{\tilde{\varepsilon}_x^2} + \overline{\tilde{\varepsilon}_x \partial_x \tilde{\varepsilon}_x}\delta x + \frac{1}{2}\overline{\tilde{\varepsilon}_x \partial_x^2 \tilde{\varepsilon}_x}\delta x^2 + \frac{1}{6}\overline{\tilde{\varepsilon}_x \partial_x^3 \tilde{\varepsilon}_x}\delta x^3 +$$

$$\frac{1}{24}\overline{\tilde{\varepsilon}_x \partial_x^4 \tilde{\varepsilon}_x}\delta x^4 + o(\delta x^4). \tag{A4}$$

The aim is to reformulate terms of the form $\overline{\tilde{\varepsilon}_x \partial_x^k \tilde{\varepsilon}_x}$ in order to specify links between the coefficients.

The variance $\overline{\tilde{\varepsilon}_x^2}$ of the normalized error field is first studied. From linearity of the expectation operator it can be deduced

that $\overline{\tilde{\varepsilon}_x^2} = \frac{1}{\sigma_x^2}\overline{\varepsilon_x^2} = 1$. Hence, the zero order term of the Taylor expansion Eq. (A4) is equal to 1.

The term $\overline{\tilde{\varepsilon}\partial_x\tilde{\varepsilon}}$ is now considered. Since $\partial_x \tilde{\varepsilon}_x^2 = 2\tilde{\varepsilon}_x \partial_x \tilde{\varepsilon}_x$, the commutation rule $\partial_{x^i}(\overline{\cdot}) = \overline{\partial_{x^i}(\cdot)}$ and the stationarity of the normalized variance field $\overline{\tilde{\varepsilon}_x^2} = 1$ imply that $\overline{\partial_x \tilde{\varepsilon}_x^2} = \partial_x \overline{\tilde{\varepsilon}_x^2} = 0 = 2\overline{\tilde{\varepsilon}_x \partial_x \tilde{\varepsilon}_x}$ or

$$\overline{\tilde{\varepsilon}_x \partial_x \tilde{\varepsilon}_x} = 0, \tag{A5}$$

so that the first order term of the Taylor expansion Eq. (A4) is zero.

For the term $\overline{\tilde{\varepsilon}_x \partial_x^2 \tilde{\varepsilon}_x}$: by using the identity $\partial_x(\tilde{\varepsilon}_x \partial_x \tilde{\varepsilon}_x) = \partial_x \tilde{\varepsilon}_x \partial_x \tilde{\varepsilon}_x + \tilde{\varepsilon}_x \partial_x^2 \tilde{\varepsilon}_x$, and the commutation rule, we have that $\overline{\tilde{\varepsilon}_x \partial_x^2 \tilde{\varepsilon}_x} = \partial_x\left(\overline{\tilde{\varepsilon}_x \partial_x \tilde{\varepsilon}_x}\right) - \overline{\partial_x \tilde{\varepsilon}_x \partial_x \tilde{\varepsilon}_x}$. With, Eq. (A5), this simplifies into

$$\overline{\tilde{\varepsilon}_x \partial_x^2 \tilde{\varepsilon}_x} = -\overline{\partial_x \tilde{\varepsilon}_x \partial_x \tilde{\varepsilon}_x}. \tag{A6}$$

5    Identifying with Eq. (A1), the second order term of the Taylor expansion Eq. (A4) can be written as

$$g_x = \overline{\partial_x \tilde{\varepsilon}_x \partial_x \tilde{\varepsilon}_x}. \tag{A7}$$

The third order term $\overline{\tilde{\varepsilon}_x \partial_x^3 \tilde{\varepsilon}_x}$ is reformulated as follows: $\partial_x(\tilde{\varepsilon}_x \partial_x^2 \tilde{\varepsilon}_x) = \partial_x \tilde{\varepsilon}_x \partial_x^2 \tilde{\varepsilon}_x + \tilde{\varepsilon}_x \partial_x^3 \tilde{\varepsilon}_x$ implies $\overline{\tilde{\varepsilon}_x \partial_x^3 \tilde{\varepsilon}_x} = \partial_x\left(\overline{\tilde{\varepsilon}_x \partial_x^2 \tilde{\varepsilon}_x}\right) - \overline{\partial_x \tilde{\varepsilon}_x \partial_x^2 \tilde{\varepsilon}_x}$, which is transformed using Eq. (A6) into $\overline{\tilde{\varepsilon}_x \partial_x^3 \tilde{\varepsilon}_x} = -\partial_x\left(\overline{\partial_x \tilde{\varepsilon}_x \partial_x \tilde{\varepsilon}_x}\right) - \overline{\partial_x \tilde{\varepsilon}_x \partial_x^2 \tilde{\varepsilon}_x}$. But with $\partial_x(\partial_x \tilde{\varepsilon}_x \partial_x \tilde{\varepsilon}_x) = 2\partial_x \tilde{\varepsilon}_x \partial_x^2 \tilde{\varepsilon}_x$, it results that

$$\overline{\partial_x \tilde{\varepsilon}_x \partial_x^2 \tilde{\varepsilon}_x} = \frac{1}{2}\partial_x\left(\overline{\partial_x \tilde{\varepsilon}_x \partial_x \tilde{\varepsilon}_x}\right), \tag{A8}$$

and then

$$\overline{\tilde{\varepsilon}_x \partial_x^3 \tilde{\varepsilon}_x} = -\frac{3}{2}\partial_x\left(\overline{\partial_x \tilde{\varepsilon}_x \partial_x \tilde{\varepsilon}_x}\right). \tag{A9}$$

Identifying with Eq. (A1), the third order term of the Taylor expansion Eq. (A4) can be written as

$$S_x = -\frac{1}{4}\partial_x g_x. \tag{A10}$$

15    For the last fourth order term $\overline{\tilde{\varepsilon}_x \partial_x^4 \tilde{\varepsilon}_x}$, the computation of the derivative $\partial_x(\tilde{\varepsilon}_x \partial_x^3 \tilde{\varepsilon}_x) = \partial_x \tilde{\varepsilon}_x \partial_x^3 \tilde{\varepsilon}_x + \tilde{\varepsilon}_x \partial_x^4 \tilde{\varepsilon}_x$, leads to $\overline{\tilde{\varepsilon}_x \partial_x^4 \tilde{\varepsilon}_x} = \partial_x\left(\overline{\tilde{\varepsilon}_x \partial_x^3 \tilde{\varepsilon}_x}\right) - \overline{\partial_x \tilde{\varepsilon}_x \partial_x^3 \tilde{\varepsilon}_x}$. From Eq. (A9), this reads $\overline{\tilde{\varepsilon}_x \partial_x^4 \tilde{\varepsilon}_x} = -\frac{3}{2}\partial_x^2\left(\overline{\partial_x \tilde{\varepsilon}_x \partial_x \tilde{\varepsilon}_x}\right) - \overline{\partial_x \tilde{\varepsilon}_x \partial_x^3 \tilde{\varepsilon}_x}$. The last term can be deduced from the use of the homogeneity of the normalized error variance $\partial_x^4 \overline{\tilde{\varepsilon}_x^2} = 0$. Since $\partial_x^4 \tilde{\varepsilon}_x^2 = \partial_x^3[2\tilde{\varepsilon}_x \partial_x \tilde{\varepsilon}_x]$, the Leibniz rule states

$$\partial_x^4 \tilde{\varepsilon}_x^2 = 2\left(\partial_x^3 \tilde{\varepsilon}_x \partial_x \tilde{\varepsilon}_x + 3\partial_x^2 \tilde{\varepsilon}_x \partial_x^2 \tilde{\varepsilon}_x + 3\partial_x \tilde{\varepsilon}_x \partial_x^3 \tilde{\varepsilon}_x + \tilde{\varepsilon}_x \partial_x^4 \tilde{\varepsilon}_x\right). \tag{A11}$$

20  With the ensemble average, it follows that $\overline{\partial_x \tilde{\varepsilon}_x \partial_x^3 \tilde{\varepsilon}_x} = -\frac{1}{4}\overline{\tilde{\varepsilon}_x \partial_x^4 \tilde{\varepsilon}_x} - \frac{3}{4}\overline{\partial_x^2 \tilde{\varepsilon}_x \partial_x^2 \tilde{\varepsilon}_x}$. Hence,

$$\overline{\tilde{\varepsilon}_x \partial_x^4 \tilde{\varepsilon}_x} = -2\partial_x^2\left(\overline{\partial_x \tilde{\varepsilon}_x \partial_x \tilde{\varepsilon}_x}\right) + \overline{\partial_x^2 \tilde{\varepsilon}_x \partial_x^2 \tilde{\varepsilon}_x}. \tag{A12}$$

Identifying with Eq. (A1), the fourth order term of the Taylor expansion Eq. (A4) can be written as

$$K_x = \frac{1}{24}\overline{\partial_x^2 \tilde{\varepsilon}_x \partial_x^2 \tilde{\varepsilon}_x} - \frac{1}{12}\partial_x^2 g_x. \tag{A13}$$

*Author contributions.* TEXT

*Competing interests.* TEXT

*Disclaimer.* TEXT

*Acknowledgements.* This work was supported by the french national program LEFE/INSU (Étude du filtre de KAlman PAramétrique, KAPA). The authors are thankful to S. E. Cohn and an anonymous reviewer for their useful comments, suggestions and corrections. CEREA is a member of Institut Pierre-Simon Laplace (IPSL).

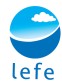

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
