# Peer review of "Parametric covariance dynamics for the nonlinear diffusive Burgers' equation"

_Nonlinear Processes in Geophysics, 2018_

## Referee Comment (RC1) · S. E. Cohn (Referee) · 31 Mar 2018

**Parametric covariance dynamics for the nonlinear diffusive Burgers equation**

**by O. Pannekoucke, M. Bocquet, and R. Ménard**

**Referee report by S. E. Cohn**

This interesting paper considers the tangent linear covariance dynamics of the (nonlinear, diffusive) Burgers equation under stochastic initial conditions. The tangent linear covariance dynamics can be expressed as a series expansion in the distance between any two points in the spatial domain. To terminate the expansion at finite order requires a closure approximation due to the diffusion term in Burgers' equation. The authors develop such an approximation through a careful examination of properties of the two-point correlation function. Their proposed approximation, appropriately termed a *locally homogeneous Gaussian closure*, terminates the series expansion at second order by approximating the fourth-order term in terms of the second-order one. This results in a system of three nonlinear, nonlinearly coupled, time-dependent PDEs to be solved on the spatial domain, for the mean, the variance, and the second-order term of the correlation function expansion. The authors carry out numerical experiments to test their approach against a Monte Carlo approach used as a basis of comparison.

The appeal of the approach considered in this paper, beyond increased scientific understanding of covariance evolution, is to dramatically reduce computational costs: one needs to solve only three PDEs instead of carrying out, for instance, the tens, hundreds, thousands or more integrations of the original (Burgers equation) dynamics with Monte Carlo approaches. The authors discuss the potential of their approach for the important applications of data assimilation and probabilistic forecasting.

The experimental results shown in the paper are convincing but also somewhat unsettling. They show that, for small initial variance (1% standard deviation), the closure approach accurately reproduces the Monte Carlo results, thus validating in this case both the closure approximation and the tangent linear covariance dynamics. For much larger initial variance (10% standard deviation), however, the closure approach reproduces the Monte Carlo results only away from the region where the gradient of the mean state steepens. Through further numerical experiments, the authors demonstrate that it is likely that it is the use of the tangent linear covariance dynamics themselves, to which the closure approximation is applied, rather than the closure approximation per se, that causes the fairly substantial difference between results of their approach and the Monte Carlo simulations in the vicinity of sharp gradients. An obvious conclusion is that further research will be needed to investigate the limitations of tangent linear covariance dynamics vis-à-vis fully nonlinear covariance dynamics, such as those obtained from Monte Carlo methods.

I did carefully check all the mathematical derivations in the paper, including those in the Appendix, and I found two, both of which could affect the experimental results. First, there is an error passing from Eq. (27b) to Eq. (28b): a coefficient 2 has crept into the third-from-last term on the right side of Eq. (28b) which does not belong there, and it is repeated in the final Eq. (29c). If this is just a typo, it simply needs to be corrected. But if this error has also made its way into the computer code, then the numerical experiments will need to be re-run.

Second, the Gaussian initial covariance function, Eq. (30), is not appropriate for the geometry of the numerical experiments. Since the domain is periodic, the distance $|x-y|$ should be replaced by a distance function that reflects this periodicity, such as the great-circle distance or chordal distance. As it stands, the covariance function has a slight first-derivative discontinuity at $|x-y|$ = D/2, and this introduces spurious odd-order terms in the series expansion of the correlation function that have been neglected. Although this might be a small effect initially, since the initial correlation length L was taken to be small, the numerical experiments showed that the correlation length grows by nearly an order of magnitude. The numerical experiments will need to be repeated with an appropriate initial covariance model.

Here are a few smaller issues, including typos:

1. It is mentioned a few times that "operator splitting" (p.2 l.25) or "time splitting" (p.6 l.26) is used in the derivation. Actually, the authors are simply carrying out the derivation term-by-term without any approximation introduced by doing so. The authors' use of this terminology is not at all standard; usually it means that a time discretization error is introduced in a numerical method. I suggest removing the terminology altogether.
2. P.7 l.13: I would change "Hilbert space" to the more general "function space" since no Hilbert space apparatus has been introduced in the paper.
3. In the title itself, the apostrophe after Burgers should be removed: the possessive is not correct here.
4. P.4 l.16: recipes → recipe
5. P.6 l.12: te → the
6. Eq. (25): One appearance of $\delta x^2$ in the second term, and one in the third term, should be removed.
7. P.10 l.8: express → expressed
8. Eq. (29): The subscript x on the symbol V should be used consistently.
9. P.13 l.33: 8.8 of → 8.8 times
10. P.14 l.9: 5.5 of → 5.5 times
11. P.14 l.12: 7.5 of → 7.5 times
12. P.18 l.13: variance field → normalized variance field
13. P.18 l.21: third term → third order term
14. P.19 l.1: fourth term → fourth order term
15. P.19 l.8: third order term → fourth order term

---

## Referee Comment (RC2) · Anonymous Referee #2 · 18 Apr 2018

[thmsa,a4paper,amsfont,12pt]article amsmath

Referee's Report on

**Parametric covariance dynamics for the nonlinear diffusive Burgers' equation**
**Olivier Pannekoucke, Marc Bocquet, and Richard Ménard**

The submitted manuscript considers a new methodology for the numerical approximation of the forecast error covariance matrix in a nonlinear setting. The methodology is applied to the nonlinear dynamics given by Burgers' equation. The results are extensions of works of the first two authors, Bocquet (2016) and Pannekoucke et al. (2016). More precisely, in Pannekoucke et al. (2016), the evolution of the error correlation function is approximated using a second order Taylor expansion with the resulting methodology not being able to reproduce the complexity of the Burgers dynamics. In this work, the authors use a fourth order Taylor expansion which much better results. The paper benefits from a nice background introduction on the uncertainty propagation and covariance dynamics. The numerical simulations presented illustrate the ability of the parametric dynamics to reproduce the main features of the true covariance dynamics emerging from a forecast Monte Carlo experiment.

The paper is very well written and the results as well as the numerical work are interesting and with potentially strong consequences to data assimilation. As such it is my recommendation that the paper is accepted for publication in the Nonlinear Processes in Geophysics journal. I have a list of minor comments/misprints that the authors should take into account when submitting the final version:

○ page 1 Abstract: I am not sure if you can say that "this study extends the PFF to nonlinear dynamics". Rather, It is a required preliminary step.

○ page 2 line 20 "numerical test*s*

○ page 3 line 2 "*B*ackground on the"

○ page 4 line 15 I am not really keen to use a semigroup in order to define a multivariate matrix. What stops you writing the matrix explicitly ?

○ page 5 line 10 "meaningful"

○ page 5 line 13 The dimension of the space plays no role in here.

○ page 6 line 5 Can I suggest that you avoid using the term "nonlinear Kalman filter". The Kalman filter is, by definition, linear. Yes, there are extensions of the Kalman filter methodology to nonlinear frameworks (the extended Kalman Filter, the ensemble Kalman filter, etc), but I believe that here you are referring to the framework which is either nonlinear or linear (Kalman).

○ page 6 line 12 "deduced from the difference"

○ page 7 line 21 As I understand it, $\overline{\partial_x \varepsilon^2}$ is the expectation of $\partial_x \varepsilon^2$, not the approximation obtained by the ensemble average.

○ page 9 line 13 There is a comma missing after the exponential.

○ page 10 line 8 "expressed"

○ page 10 line 23 How large is the ensemble for the numerical experiment ? Why do you call it nonlinear ?

○ page 11 Caption for Figure 1: the figure has 6 curves. I assume you plot the solution at 0.8T too (same for Figure 4 page 15, Figure 5 page 16).

○ page 13 line 17 What does the 17.6% figure signify ?

○ page 14 Figure 3 What is the maximum difference between the two graphs ?

○ page 15 line 10 In the case of Burgers' equation, can you provide an estimate of the length of the time interval throughout which the method presented in the paper gives meaningful results.

○ page 16 Figure 16. Again, it would help to know how large is the ensemble used in the numerical experiments.

○ page 17 line 22 "The aim of this section"

○ page 18 line 2,4 Something went wrong with the display of the formulae (A3) and (A4)

---

## Author Comment (AC1) · 29 May 2018

We would like to thank S.E. Cohn for his review on our paper and for giving us the opportunity to improve our paper.

We have improve the description of the numerical experiments with some details on the implementation used: finite difference for the spacial discretization, a fourth order Runge-Kutta for the time scheme and we have specified the numerical setting (time step, numerical value of the diffusion coefficient). The ensemble size has been increased to 6400 in order to limit the sampling noise, and a single ensemble of normalized error has been generated then used with appropriate initial error magnitude – this reduces the sampling fluctuations when comparing the numerical results from a method to another.

In order to investigate the limitation of the tangent-linear covariance dynamics the manuscript incorporates new results (even if further research are still needed to investigation more completely the limitations of the parametric formulation as highlighted in the manuscript):

A study of the mean predicted by the parametric model and estimated from the ensemble has been introduced in order to illustrate the ability of the PKF to provide an estimation of the true mean state when small non-linearities are present: see Fig. 4 and the new section 4.3.1. A long term experiment has been introduced to determine if there is an exponential growth of the error that could be a side effect of the tangent-linear approximation: see Fig. 8 and the new section 4.3.3. The discussion of the results has been put in a new section 4.3.4.

We copied your commentary in italics below, we reply in normal blue font

Major comments:

two errors:
1) *"First, there is an error passing from Eq. (27b) to Eq. (28b): a coefficient 2 has crept into the third-from-last term on the right side of Eq. (28b) which does not belong there, and it is repeated in the final Eq. (29c). If this is just a typo, it simply needs to be corrected. But if this error has also made its way into the computer code, then the numerical experiments will need to be re-run."*

This has been corrected, thank you very much.

2) *"Second, the Gaussian initial covariance function, Eq. (30), is not appropriate for the geometry of the numerical experiments. Since the domain is periodic, the distance |x-y| should be replaced by a distance function that reflects this periodicity, such as the great-circle distance or chordal distance. As it stands, the covariance function has a slight first-derivative discontinuity at |x-y| = D/2, and this introduces spurious odd-order terms in the series expansion of the correlation function that have been neglected. Although this might be a small effect initially, since the initial correlation length L was taken to be small, the numerical experiments showed that the correlation length grows by nearly an order of magnitude. The numerical experiments will need to be repeated with an appropriate initial covariance model."*

For the experiment considered in the manuscript, only the initial covariance Eq.(30) (from the previous version) is needed. Since the length-scale considered is relatively small considering the length of the domain, the Gaussian correlation applies here. However, we agree with the referee that this is not strictly a valid correlation function, and a chordal distance has been introduced in accordance with the geometry of the domain. This modification does not change the results but is theoretically better.

Minor comments and typos:

1. *It is mentioned a few times that "operator splitting" (p.2 l.25) or "time splitting" (p.6 l.26) is used in the derivation. Actually, the authors are simply carrying out the derivation term-by-term without any approximation introduced by doing so. The authors' use of this terminology is not at all standard; usually it means that a time discretization error is introduced in a numerical method. I suggest removing the terminology altogether.*
We agree with the referee that the terminology "time-splitting" could introduce a confusion with the classical numerical time-splitting. In order to avoid this confusion, we have introduced the following lines in section 3.1, where the splitting method is mentioned:

*"The splitting strategy is a theoretical method to deduced the so-called infinitesimal generator of an evolution equation, by taking advantage of the Lie-Trotter formula to separate each processes (or appropriate arrangements of the processes). This strategy should not be confused with the numerical time-splitting which introduces numerical errors (Sportisse, 2007)."*

2. *P.7 l.13: I would change "Hilbert space" to the more general "function space" since no Hilbert space apparatus has been introduced in the paper.*
Yes the Hilbert structure is not important here and it has been replace by "function space" as suggested by the referee.

3. *In the title itself, the apostrophe after Burgers should be removed: the possessive is not correct here.*
The typos is now corrected.

4. *P.4 l.16: recipes → recipe*
The typos is now corrected.

5. *P.6 l.12: te → the*
The typos is now corrected.

6. *Eq. (25): One appearance of $\delta x$ 2 in the second term, and one in the third term, should be removed.*
The Taylor expansion Eq.(25) has been corrected.

7. *P.10 l.8: express → expressed*
The typos is now corrected.

8. *Eq. (29): The subscript $x$ on the symbol $V$ should be used consistently.*
The subscript x has been removed from Eq.(29).

9. *P.13 l.33: 8.8 of → 8.8 times*
The typos is now corrected.

10. *P.14 l.9: 5.5 of → 5.5 times*
The typos is now corrected.

11. *P.14 l.12: 7.5 of → 7.5 times*
The typos is now corrected.

*12. P.18 l.13: variance field → normalized variance field*
The typos is now corrected.

*13. P.18 l.21: third term →  third order term*
The typos is now corrected.

*14. P.19 l.1: fourth term →  fourth order term*
The typos is now corrected.

*15. P.19 l.8: third order term →  fourth order term*
The typos is now corrected.

---

## Author Comment (AC3) · 29 May 2018

**Reply to rev. 2**

We would like to thank the referee for his/her review on our paper and for giving us the opportunity to improve our paper.

We have improve the description of the numerical experiments with some details on the implementation used: finite difference for the spacial discretization, a fourth order Runge-Kutta for the time scheme and we have specified the numerical setting (time step, numerical value of the diffusion coefficient). The ensemble size has been increased to 6400 in order to limit the sampling noise, and a single ensemble of normalized error has been generated then used with appropriate initial error magnitude – this reduces the sampling fluctuations when comparing the numerical results from a method to another.

In order to investigate the limitation of the tangent-linear covariance dynamics the manuscript incorporates new results (even if further research are still needed to investigation more completely the limitations of the parametric formulation as highlighted in the manuscript):

A study of the mean predicted by the parametric model and estimated from the ensemble has been introduced in order to illustrate the ability of the PKF to provide an estimation of the true mean state when small non-linearities are present: see Fig. 4 and the new section 4.3.1. A long term experiment has been introduced to determine if there is an exponential growth of the error that could be a side effect of the tangent-linear approximation: see Fig. 8 and the new section 4.3.3. The discussion of the results has been put in a new section 4.3.4.

We copied your commentary in italics below, we reply in normal blue font

**1) page 1 Abstract: "Abstract: I am not sure if you can say that "this study extends the PKF to nonlinear dynamics". Rather, It is a required preliminary step."**

Since the parametric model is designed to the nearly nonlinear dynamics we think this terminology is appropriate. This is supported by the introduction of the mean predicted by the ensemble versus the parametric model (Fig. 4) where the mean is different from the reference state at T.

2) page 2 line 20 "numerical tests" The typos is now corrected

3) *page 3 line 2 "Background on the*" The typos is now corrected

4) page 4 line 15 I am not really keen to use a semigroup in order to define a multivariate matrix. What stops you writing the matrix explicitly ? The matrix results from numerical integration of the diffusion equation, it is not easy to write the matrix explicitly.

5) *page 5 line 10 "meaningful"* The typos is now corrected.

**6) page 5 line 13 The dimension of the space plays no role in here.**

We agree with the referee comment but we think it is interesting to extrapolate the 1D situation to the 2D/3D case much important for further applications, where Eq. (11) offers a systematic derivation of the dynamics, whatever the dimension is.

7) page 6 line 5 Can I suggest that you avoid using the term "nonlinear Kalman filter".

The Kalman filter is, by definition, linear. Yes, there are extensions of the Kalman filter methodology to nonlinear frameworks (the extended Kalman Filter, the ensemble Kalman filter, etc), but I believe that here you are referring to the framework which is either nonlinear or linear (Kalman).

The terminology has been employed by Cohn (1993) and corresponds to the next order of the extended Kalman filter where the magnitude of the fluctuation influences the mean state. The sentence has been rephrase as follows:

" Note that the fluctuation-mean flow interaction leads to the

Gaussian second-order filter \citep[sec. 9.3]{Jazwinski1970book}, and is important in nonlinear Kalman-like filters \citep{Cohn1993MWR}."

8) *page 6 line 12 "deduced from the difference"* The typos is now corrected.

9) page 7 line 21 As I understand it,  $\langle e^2 \rangle$  is the expectation of  $\partial x \varepsilon 2$ , not the approximation obtained by the ensemble average."

Yes this is correct: in this expression there is no approximation, and you are right this corresponds to the expectation operator of the derivative of  $\epsilon^2$ . At a theoretical level, the "ensemble average" considering an infinite ensemble size is equivalent to the expectation operator. And this infinite ensemble average should not be confused with the finite ensemble average as used with an EnKF: the theoretical derivation does not rely on any finite ensemble as encountered in the EnKF. The deterministic equations Eq.(29) of the parametric model does not need any ensemble.

10) *page* 9 *line* 13 *There is a comma missing after the exponential.* The typos is now corrected.

11) *page 10 line 8 "expressed"*. The typos is now corrected.

12) page 10 line 23 How large is the ensemble for the numerical experiment ? Why do you call it nonlinear ?

The ensemble size was mentioned in p11, line 12: 1600 members. The sentence has been rephrased as: *"Then, the PKF is assessed using a large ensemble of nonlinear forecasts (6400 members)"*.

13) page 11 Caption for Figure 1: the figure has 6 curves. I assume you plot the solution at 0.8T too (same for Figure 4 page 15, Figure 5 page 16) : The time 0.8T has been added in the caption.

14) page 13 line 17 What does the 17.6% figure signify ? This refers to the result at time t=T. The sentence has been rephrase to clarify: "with a low relative error  $||K - K^{GC}||/|K|| \le 0 \le 0.4$ " (respectively 17.6) at time  $0 \le 0.5$  (respectively T), "

15) *page 14 Figure 3 What is the maximum difference between the two graphs ?* We have precise the maximum difference of the two kurtosis normalized by K\_G: 0.05

16) page 15 line 10 In the case of Burgers' equation, can you provide an estimate of the length of the time interval throughout which the method presented in the paper gives meaningful results.

In order to improve this point, additional results have been incorporated in the manuscript: -1- Prediction of the mean:

For the Burgers equation, in the parametric model Eq.(29), the dynamics of the mean flow is

the true dynamics of the ensemble mean. We have insisted on the exactness of Eq.(14a) to predict the ensemble mean with the sentence: "Moreover, as pointed out in Ménard (1994), Eq. (14a) is the exact the ensemble mean for the Burgers dynamics, while Eq. (14b) is an approximated dynamics. As a consequence, if the variance field is the true one, then the mean predicted by Eq. (14a) is the true ensemble mean (Ménard, 1994, sec. 5.5.2)." (sec. 3.1)

The new section 4.3.1 illustrates the case of the mean (see also the new Fig. 4).

-2- Introduction of a long term experiment:

The new Fig. 8 and new section 4.3.4 have been introduce to tackle (but only in part) the time where the parametric model is valid.

**17) page 16 Figure 16. Again, it would help to know how large is the ensemble used in the numerical experiments.**

In the new version of the manuscript, the ensemble size has been increased to 6400 with a single ensemble of normalized error to facilitate the comparison of the numerical experiments.

18) *page 17 line 22 "The aim of this section"* The typos is now corrected.

19) *page 18 line 2,4 Something went wrong with the display of the formulae (A3) and (A4)* This is due to the "manuscript version" configuration of the manuscript (single column while "article version" is two columns). The two column version of the manuscript does not present this wrong display.